# EvReflection: Event-Driven Micro-Dynamics for Reflection Removal

**Jiaxiao Wang** [1]  **Dachun Kai** [1]  **Huyue Zhu** [1]  **Quanquan Hu** [1]  **Zhenyang Xu** [1]  **Xiaoyan Sun** [1 2]

## Abstract

Despite remarkable progress in reflection removal, current methods primarily exploit static image priors from a single frame and still suffer from severe residual artifacts due to the inherent ambiguity between the reflection and transmission layers. In this paper, we propose leveraging event signals to break this ambiguity. By employing event cameras to capture micro-dynamics, we reveal the differential motion between these two layers. We thereby present a novel event-driven reflection removal network, EvReflection, that utilizes these dynamic cues for layer separation. Specifically, we design a Micro-Dynamics Decoupler to disentangle layer-specific motions from event streams as priors, which then guide a Parallax-Attention Rectifier to cleanly remove artifacts from the RGB image. Furthermore, to address data scarcity, we develop a parallax-aware simulation pipeline and construct the $EVR^2$ benchmark dataset, the first real-world dataset for this task. Extensive experiments demonstrate that EvReflection achieves state-of-the-art performance on both synthetic and real-world benchmarks, surpassing the best competing method by more than 1.6 dB and 1.2 dB in PSNR, respectively. The code, dataset, and pre-trained models are available at https://github.com/JiaxiaoWang/EvReflection.

## 1. Introduction

Images captured through transparent media, such as glass windows, often suffer from reflections that mix with the background transmission. This visual degradation obstructs primary content and hinders downstream tasks like object detection (Cao et al., 2023). Consequently, reflection removal is a fundamental task with critical applications in photography, surveillance, and robotics.

[1]University of Science and Technology of China [2]Institute of Artificial Intelligence, Hefei Comprehensive National Science Center. Correspondence to: Dachun Kai <dachunkai@mail.ustc.edu.cn>.

*Proceedings of the 43rd International Conference on Machine Learning*, Seoul, South Korea. PMLR 306, 2026. Copyright 2026 by the author(s).

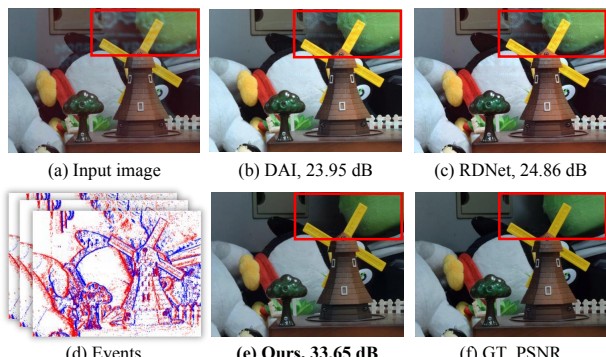

*Figure 1.* Visual comparison in a highly reflective scene. Existing methods like DAI (Hu et al., 2026) and RDNet (Zhao et al., 2025) still leave visible artifacts due to the inherent ambiguity between layers. By leveraging event signals (d) to capture micro-dynamics, our method successfully removes reflections and recovers a clean image with a significant PSNR gain (+9.70 dB over DAI).

Despite its importance, reflection removal remains a highly ill-posed problem. Existing methods generally fall into two categories: single-image and multi-frame approaches. Single-image methods (Zhu et al., 2024; Hu et al., 2026) rely on priors to distinguish layers but fail when reflections mimic background patterns, leading to inherent ambiguity and severe artifacts, as shown in Figure 1. Conversely, multi-frame methods (Zhang et al., 2024; He et al., 2025) utilize motion parallax but typically require significant camera movement (large baseline) to generate sufficient optical flow. This limits their practicality in scenarios where large motions are constrained or unavailable.

To break this ambiguity without relying on large-scale motion, we introduce event signals. Unlike frame-based cameras, event sensors record brightness changes with microsecond resolution (Gallego et al., 2020). Our key insight is to leverage subtle camera motion. Since the reflection (glass surface) and background lie at different depths, they exhibit distinct motion patterns even under slight movement, as illustrated in Figure 2. Event cameras precisely capture these *micro-dynamics*, providing physically grounded cues to disentangle the layers that conventional frame-based cameras often miss due to limited frame rates or resolution.

Accordingly, we present **EvReflection**, an event-driven network designed to exploit these cues. We propose a Micro-

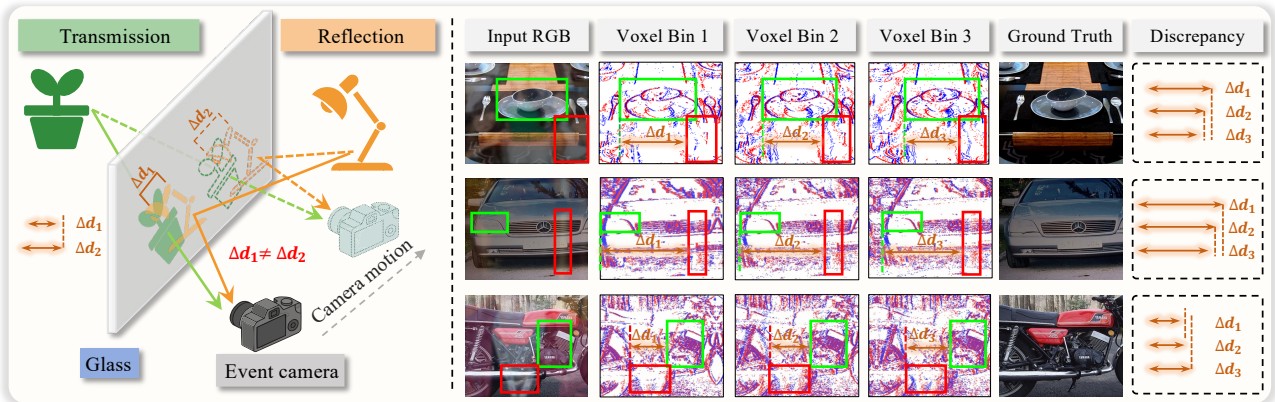

(a) Micro-dynamics via motion parallax    (b) Micro-dynamics in event streams

*Figure 2.* Illustration of event-driven micro-dynamics via motion parallax. (a) Since the transmission and reflection layers reside at different depths, even subtle camera motion induces distinct pixel displacements between the two layers. (b) Event voxel slices reveal this phenomenon: the spatial offset $\Delta d$ between transmission (green) and reflection (red) layers differs across temporal bins.

Dynamics Decoupler (MDD) to disentangle layer-specific motions from event streams as texture-agnostic dynamic priors. These priors then drive a Parallax-Attention Rectifier (PAR) to accurately distinguish deceptive reflection artifacts from background textures in the RGB domain. By fusing the high temporal precision of events with RGB semantic context, our method achieves robust layer separation.

However, event-based reflection removal is primarily hindered by the scarcity of paired RGB-event data. To bridge this gap, we develop a parallax-aware simulation pipeline modeling optical geometry and 3D camera trajectories to synthesize physically consistent data. Furthermore, we collect the $\mathbf{EVR}^2$ (**EV**ent-based **R**eflection **R**emoval) benchmark dataset to evaluate generalization across real-world environments. Extensive experiments demonstrate that EvReflection significantly outperforms existing methods, restoring clean transmission layers even in challenging scenarios.

The main contributions of this paper are as follows:

- We propose a novel perspective for reflection removal by introducing event signals, leveraging micro-dynamics induced by subtle camera motion to break the reflection-background ambiguity.

- We develop EvReflection, incorporating MDD and PAR modules. MDD disentangles layer-specific motions from event streams as dynamic priors, which are then leveraged by PAR to spatially modulate RGB features for accurate artifact removal.

- We design a parallax-aware simulation pipeline to address data scarcity and construct the $\mathbf{EVR}^2$ benchmark dataset. Experiments demonstrate superior performance in both synthetic and real-world scenarios.

## 2. Related Work

**Single-image Reflection Removal.** Early single-image methods relied on handcrafted priors, such as gradient sparsity (Chung et al., 2009; Li et al., 2025b) and ghosting cues (Shih et al., 2015), to constrain the ill-posed decomposition. Recent deep learning approaches leverage CNNs or GANs to learn the mapping from mixed images to clean backgrounds directly, sometimes utilizing semantic, edge, or polarization information for guidance (Song et al., 2023; Deng et al., 2026). Despite these advancements, single-image methods rely solely on spatial appearance priors and lack complementary cues to resolve the inherent ambiguity, leading to residual artifacts or incorrect removal.

**Multi-image Reflection Removal.** Multi-image methods (Niklaus et al., 2021; Zhang et al., 2024; He et al., 2025) address the ambiguity by exploiting motion parallax, where reflection and background layers exhibit different motions due to depth disparity. Traditional approaches align frames using optical flow or SIFT features to separate layers, while learning-based methods aggregate temporal features to enforce consistency. However, these methods typically require significant camera movement, *i.e.*, a large baseline, to generate sufficient optical flow. They often struggle in casual handheld scenarios where motion is minimal, as the lack of geometric cues hinders accurate layer separation.

**Event-based Vision.** Event cameras asynchronously record brightness changes with micro-second resolution and high dynamic range (Han et al., 2023; Xiao et al., 2024), offering significant advantages over frame-based sensors (Xu et al., 2025; Yan et al., 2025). They have been successfully applied to high-speed low-level vision tasks, such as motion deblurring (Yu et al., 2024; Yang et al., 2024; Xiao et al., 2026) and super-resolution (Kai et al., 2023; 2024; 2025;

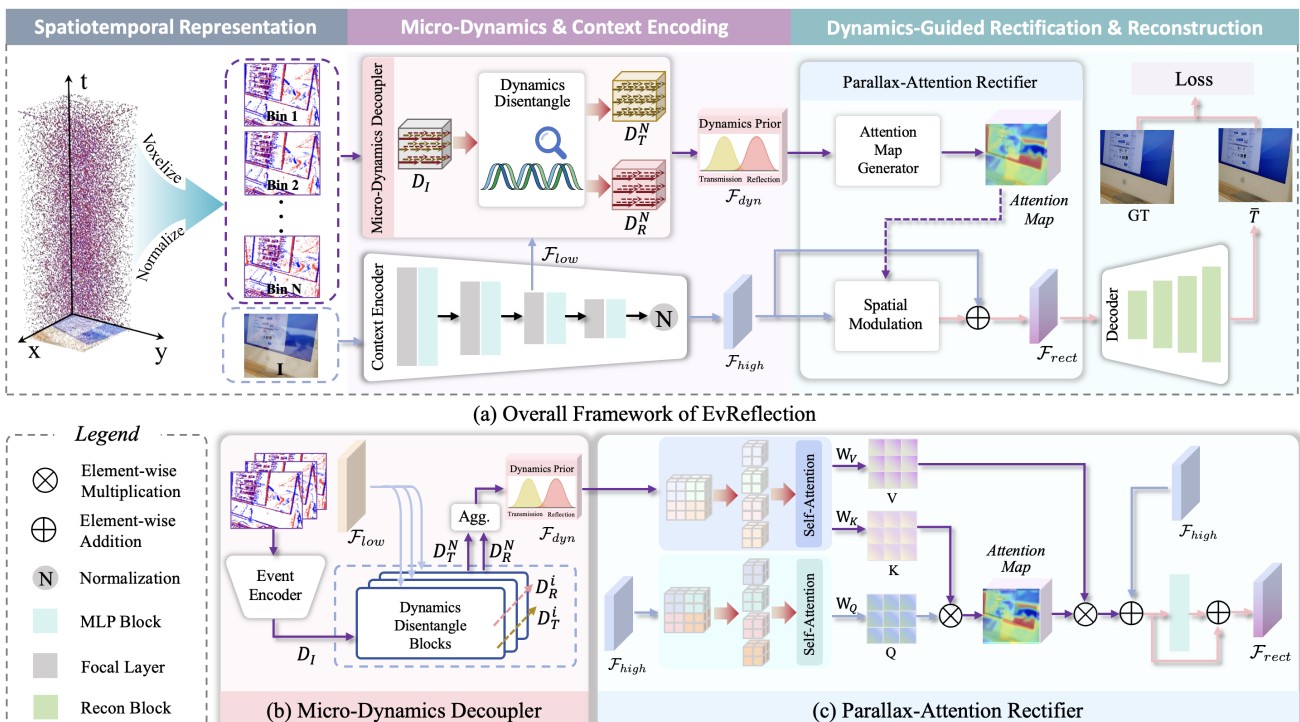

(a) Overall Framework of EvReflection

(b) Micro-Dynamics Decoupler

(c) Parallax-Attention Rectifier

*Figure 3.* Overview of the EvReflection framework. Event streams are voxelized into a spatiotemporal representation. The Micro-Dynamics Decoupler disentangles mixed event dynamics into layer-specific transmission and reflection priors. These priors are fed into the Parallax-Attention Rectifier, which leverages cross-modal attention to spatially modulate RGB features and remove reflection artifacts.

2026a;b). Fusing event signals with other modalities has also shown strong potential in related restoration tasks (Liu et al., 2024; 2025; Li et al., 2024; 2025a). Inspired by this, we leverage event cameras to capture micro-dynamics induced by subtle handheld motion, enabling robust layer separation even where frame-based methods fail.

## 3. Method

In this section, we first formulate the reflection removal problem from an event-based perspective, proving the theoretical separability of layers based on micro-dynamics. We then present EvReflection, a novel framework that translates this theory into a deep learning model. The overall architecture is illustrated in Figure 3.

### 3.1. Problem Formulation and Analysis

Event cameras operate asynchronously, triggering events whenever the logarithmic change in intensity $\mathcal{I}$ at a pixel exceeds a threshold (Gallego et al., 2020). For a microsecond-level window, the event stream can be approximated as a continuous signal $\mathcal{E}(\mathbf{x}, t) \approx \partial \ln \mathcal{I}(\mathbf{x}, t)/\partial t$.

We model the image formation as a linear superposition $\mathcal{I} = \mathcal{T} + \mathcal{R}$, where $\mathcal{T}$ and $\mathcal{R}$ denote the latent transmission and reflection layers, respectively. Assuming brightness

constancy, the temporal derivative relates to spatial gradients via the Optical Flow Constraint Equation (Horn & Schunck, 1981; Benosman et al., 2013). Substituting this into the event model yields the Event-Gradient Constraint:

$$\mathcal{I} \cdot \mathcal{E} = -\left(\langle \mathbf{u}_T, \nabla \mathcal{T}\rangle + \langle \mathbf{u}_R, \nabla \mathcal{R}\rangle\right), \quad (1)$$

where $\mathbf{u}_T, \mathbf{u}_R$ denote the layer-specific motion vectors, and $\nabla \mathcal{T}, \nabla \mathcal{R}$ are the corresponding spatial gradients.

To solve for the latent gradients, we aggregate $N$ observations within a local spatiotemporal neighborhood. Assuming the motion vectors share a local unit direction $\mathbf{d}$ with varying scalar magnitudes $v_T$ and $v_R$ (i.e., $\mathbf{u}_k = v_k \mathbf{d}$), we construct a linear system $\mathbf{A}\mathbf{x} = \mathbf{b}$:

$$\underbrace{\begin{bmatrix} v_T(t_1) & v_R(t_1) \\ \vdots & \vdots \\ v_T(t_N) & v_R(t_N) \end{bmatrix}}_{\mathbf{A} \in \mathbb{R}^{N \times 2}} \underbrace{\begin{bmatrix} x_T^{\parallel} \\ x_R^{\parallel} \end{bmatrix}}_{\mathbf{x}} = -\underbrace{\begin{bmatrix} \mathcal{I}_{t_1}\mathcal{E}_{t_1} \\ \vdots \\ \mathcal{I}_{t_N}\mathcal{E}_{t_N} \end{bmatrix}}_{\mathbf{b}}, \quad (2)$$

where $x_T^{\parallel}, x_R^{\parallel}$ are the spatial gradients projected onto $\mathbf{d}$. The separation problem is then formulated as minimizing the residual $\|\mathbf{A}\mathbf{x} - \mathbf{b}\|^2$.

**Theoretical Solvability.** The closed-form solution depends on the Hessian matrix $\mathbf{H} = \mathbf{A}^\top \mathbf{A}$. A critical insight is that $\mathbf{H}$ becomes positive definite if and only if the columns of

**A** are linearly independent. This condition holds strictly when there is motion parallax, *i.e.*, $v_T(t) \neq v_R(t)$. This theoretically confirms that distinct micro-dynamics induced by subtle camera shakes guarantee a unique solution, effectively resolving the ill-posedness inherent in single-image methods (Szeliski et al., 2000; Gai et al., 2011).

Guided by this proof, we design EvReflection to approximate this inverse process, utilizing a deep network to learn the mapping from event micro-dynamics to clean transmission layers without explicit optimization.

### 3.2. Overview of EvReflection

As shown in Figure 3(a), EvReflection takes a mixed RGB image **I** and a concurrent raw event stream as input. To leverage the rich temporal information of events, we first transform the asynchronous stream into a high-resolution spatiotemporal voxel grid $\mathbf{V} \in \mathbb{R}^{B \times H \times W}$ (Zhu et al., 2019), preserving micro-second motion details.

The framework operates in two cooperative stages. First, the MDD analyzes the event voxel grid to disentangle mixed motion, producing layer-specific dynamic features. Second, the PAR utilizes these features as priors to explicitly guide RGB restoration. Unlike standard concatenation-based fusion, PAR uses dynamic cues to spatially attend to and distinguish reflection artifacts within the RGB context.

### 3.3. Micro-Dynamics Decoupler

The MDD is specifically engineered to extract layer-specific motion patterns from the input spatiotemporal voxel grid **V**. As illustrated in Figure 3(b), this module functions through a combination of hierarchical spatial encoding and an iterative recurrent disentanglement process, which allows for the progressive separation of mixed dynamic signals.

**Feature Encoding.** We first employ a lightweight Event Encoder **E**, comprising five residual blocks (Wang et al., 2018), to transform the sparse, asynchronous voxel input **V** into a high-dimensional dense feature space. This process yields the initial dynamics embedding $D_I = \mathbf{E}(\mathbf{V})$. By projecting raw events into a continuous feature manifold, this stage distills low-level edge motion cues from the temporal stream while suppressing inherent stochastic sensor noise.

**Dual-Branch Recurrent Disentanglement.** To resolve the fundamental ambiguity between layers, we introduce a stack of Dynamics Disentangle Blocks (DDB). Motivated by the physical constraint that transmission and reflection follow distinct trajectories ($v_T \neq v_R$), we design a dual-branch architecture that maintains two evolving feature states: transmission dynamics $D_T^i$ and reflection dynamics $D_R^i$ at the $i$-th iteration. We initialize these states as $D_T^0 = D_R^0 = D_I$ to establish a shared motion prior. Crucially, to anchor the disentanglement process with reliable structural informa-

tion, we inject the low-level RGB feature $\mathcal{F}_{low}$ (extracted from the shallow layers of the Context Encoder) into each recurrent unit. These features provide high-frequency spatial boundaries that guide the event-driven motion estimation. The dual-stream update rule is formulated as:

$$
\begin{aligned}
D_T^i &= \text{DDB}\left(D_T^{i-1}, [D_I, \mathcal{F}_{low}, D_R^{i-1}]\right), \\
D_R^i &= \text{DDB}\left(D_R^{i-1}, [D_I, \mathcal{F}_{low}, D_T^{i-1}]\right),
\end{aligned}
\tag{3}
$$

where $[\cdot]$ denotes the concatenation operation. This interactive recurrence allows the network to refine layer separation by considering competing motion cues of both layers.

**Prior Aggregation.** After $N$ iterations, decoupled features are fused via a motion-aware aggregation unit to generate the final texture-agnostic dynamic prior: $\mathcal{F}_{dyn} = \text{Conv}([D_T^N, D_R^N])$. Unlike raw RGB features suffering from visual blending, $\mathcal{F}_{dyn}$ strictly encodes structural motion boundaries of both layers, providing clean and physically grounded guidance for the subsequent rectification stage.

### 3.4. Parallax-Attention Rectifier

While MDD successfully separates motion patterns, restoring a visually pleasing image requires a sophisticated mechanism to integrate dynamic cues with rich photometric details. The PAR module, illustrated in Figure 3(c), bridges this gap through a dynamics-guided cross-modal attention mechanism designed to purify the corrupted feature space.

**Context Encoding.** We employ a pre-trained Focal-Net (Yang et al., 2022) as the Context Encoder to extract hierarchical semantic features from the mixed RGB image **I**. To ensure multi-scale synergy, we utilize features from distinct stages: the shallow feature $\mathcal{F}_{low}$ provides structural anchors for the MDD, while the deep feature $\mathcal{F}_{high}$ encapsulates abstract semantic context, serving as the primary substrate for reconstruction.

**Attention Map Generation.** To precisely locate and isolate reflection artifacts, we formulate the cross-modal alignment as a query-key matching problem. First, both the visual features $\mathcal{F}_{high}$ and dynamics priors $\mathcal{F}_{dyn}$ are enhanced via independent Self-Attention (SA) blocks to capture global contextual dependencies and long-range structural correlations. We then project the enhanced visual features into a query (**Q**) and the dynamics priors into a key (**K**) via learnable weight matrices $\mathbf{W}_Q$ and $\mathbf{W}_K$, respectively. The spatial Attention Map **M** is computed by measuring the pixel-wise similarity:

$$
\mathbf{M} = \text{Softmax}\left(\frac{\mathbf{Q}\mathbf{K}^\top}{\sqrt{d}}\right),
\tag{4}
$$

where $d$ is the feature dimension scaling factor. Physically, **M** acts as a soft mask: high attention scores indicate regions where local visual textures align with the transmission

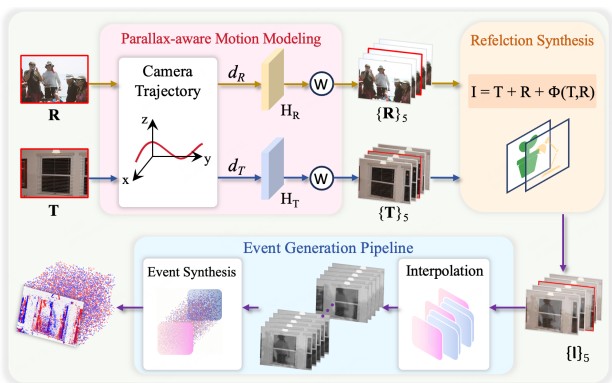

*Figure 4.* Parallax-aware simulation pipeline. Given transmission and reflection images, we model 3D camera trajectories to simulate physically accurate motion parallax, producing blended image sequences that are then upsampled and converted to synchronized event streams via the ESIM (Rebecq et al., 2018) simulator.

micro-dynamics, while low scores highlight areas dominated by reflection interference.

**Spatial Modulation and Reconstruction.** The core rectification is achieved by projecting the dynamics prior into a Value (**V**) embedding using $\mathbf{W}_V$. This module then spatially modulates the RGB context by performing element-wise multiplication with the attention map:

$$\mathcal{F}_{rect} = \mathcal{F}_{high} + \mathbf{M} \cdot \mathbf{V}. \tag{5}$$

This operation suppresses reflection artifacts while preserving transmission details. The rectified features are refined by a Feed-Forward Network and decoded to yield the clean transmission $\bar{\mathbf{T}}$.

## 4. Experiments

In this section, we describe our data collection and generation, followed by implementation details. We then demonstrate the superiority of EvReflection through extensive comparisons with state-of-the-art methods and validate the effectiveness of our core modules via detailed ablation studies.

### 4.1. Datasets and Simulation Pipeline

Capturing pixel-perfect ground truth (*i.e.*, clean transmission and reflection layers) alongside high-speed event streams in the real world is extremely challenging. To bridge this data gap, we developed a parallax-aware simulation pipeline to generate large-scale synthetic data and collected a dedicated real-world dataset to test our model's generalization.

**Parallax-aware Simulation Pipeline.** Figure 4 shows that we synthesize realistic event sequences by modeling optical geometry and camera kinematics. Using high-quality images from PASCAL VOC (Everingham et al., 2010) and SIR$^2$ (Wan et al., 2017) as source layers, we simulate physi-

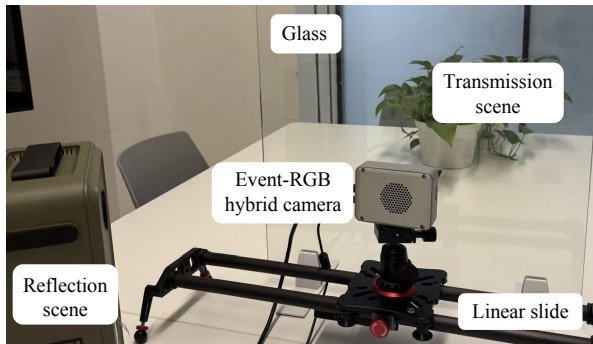

*Figure 5.* Real-world data acquisition setup for the EVR$^2$ dataset. An Event-RGB hybrid camera is mounted on a motorized linear slide, which ensures smooth and repeatable horizontal translation to induce consistent motion parallax across all recorded scenes.

cally accurate micro-dynamics by modeling camera movement in 3D space. We assign virtual depths, $d_T$ and $d_R$, to the transmission and reflection layers ($d_T \neq d_R$). Following a smooth, random camera trajectory $\mathcal{C}(t)$, we compute homography matrices $\mathbf{H}_T(t)$ and $\mathbf{H}_R(t)$ for each step. Static images are then warped to produce a video sequence:

$$\{T_t\}_{t=1}^5 = \mathcal{W}(T, \mathbf{H}_T(t)), \quad \{R_t\}_{t=1}^5 = \mathcal{W}(R, \mathbf{H}_R(t)) \tag{6}$$

where $\mathcal{W}(\cdot)$ is the perspective warping operation. The mixed sequence $\{I_t\}$ is then blended using the physically grounded model from DSRNet (Hu & Guo, 2023): $I_t = T_t + R_t + \Phi(T_t, R_t)$, where $\Phi$ represents potential non-linearities.

Since directly converting low-frame-rate video to events can cause temporal aliasing, we first use a video frame interpolation network to upsample $\{I_t\}$ to a high frame rate. Finally, we process the interpolated video through the ESIM simulator (Rebecq et al., 2018) to generate the asynchronous event stream $\mathcal{E}$. This process ensures our synthetic events maintain the microsecond resolution and accurate polarity changes consistent with real-world sensor dynamics.

**The EVR$^2$ Benchmark Dataset.** To bridge the gap between simulation and reality, we introduce the **EVR$^2$** dataset, the first real-world benchmark dataset for this task. As shown in Figure 5, we constructed a precision capture system featuring a high-resolution Event-RGB hybrid camera[1] ($608 \times 768$) mounted on a motorized linear slide. This setup ensures smooth and consistent horizontal translation, inducing reliable motion parallax essential for validating layer separation in complex scenarios.

To study the complex effects of refractive geometry, we recorded 140 distinct scenes using glass plates of three specific thicknesses: 3mm, 5mm, and 8mm. This yielded a

---

[1] We utilize the Shimeta Lingguang-1 camera equipped with the AlpsenTek ALPIX-Eiger sensor, which captures natively synchronized RGB frames and event streams.

*Table 1.* Quantitative comparison on the SIR$^2$ benchmark (Wan et al., 2017). We compare EvReflection with state-of-the-art single-image (S) and multi-image (M) based methods, where our method additionally leverages event cues (S+E). **Red** and blue indicate the best and second-best performance, respectively. Missing entries indicate unavailable source code.

| Methods | Type | Objects (200) | | | Postcards (199) | | | Wild (55) | | | Average | | |
|---|---|---|---|---|---|---|---|---|---|---|---|---|---|
| | | PSNR↑ | SSIM↑ | LPIPS↓ | PSNR↑ | SSIM↑ | LPIPS↓ | PSNR↑ | SSIM↑ | LPIPS↓ | PSNR↑ | SSIM↑ | LPIPS↓ |
| IBCLN (Li et al., 2020) | S | 24.87 | 0.893 | 0.100 | 23.39 | 0.875 | 0.163 | 24.71 | 0.886 | 0.132 | 24.20 | 0.884 | 0.131 |
| YTMT (Hu & Guo, 2021) | S | 24.87 | 0.896 | 0.092 | 22.91 | 0.884 | 0.145 | 25.48 | 0.890 | 0.116 | 24.08 | 0.890 | 0.118 |
| LANet (Dong et al., 2021) | S | 24.36 | 0.898 | 0.089 | 23.72 | 0.903 | 0.117 | 25.48 | 0.890 | 0.096 | 24.08 | 0.890 | 0.102 |
| SOLD (Liu et al., 2021) | M | 24.13 | 0.909 | 0.074 | 18.75 | 0.886 | 0.176 | 25.88 | 0.904 | 0.119 | 21.98 | 0.898 | 0.124 |
| PNACR (Wang et al., 2023) | S | 24.73 | 0.897 | – | 23.11 | 0.890 | – | 25.69 | 0.903 | – | 24.14 | 0.895 | – |
| DSRNet (Hu & Guo, 2023) | S | 26.74 | 0.920 | 0.069 | 24.83 | 0.911 | 0.119 | 26.11 | 0.906 | 0.113 | 25.83 | 0.914 | 0.096 |
| RRW (Zhu et al., 2024) | S | 26.84 | 0.927 | 0.075 | 24.38 | 0.892 | 0.157 | 26.77 | 0.910 | 0.099 | 25.75 | 0.910 | 0.114 |
| L-DiffER (Hong et al., 2024) | S | – | – | – | – | – | – | – | – | – | 25.18 | 0.911 | – |
| DSIT (Hu et al., 2024) | S | 26.72 | 0.925 | 0.076 | 25.19 | 0.925 | 0.109 | 27.08 | 0.917 | 0.080 | 26.09 | 0.924 | 0.091 |
| RDNet (Zhao et al., 2025) | S | 26.78 | 0.921 | 0.068 | 26.33 | 0.922 | 0.096 | 27.84 | 0.915 | 0.082 | 26.71 | 0.921 | 0.082 |
| DAI (Hu et al., 2026) | S | 27.39 | 0.919 | 0.079 | 27.50 | 0.921 | 0.103 | 27.54 | 0.912 | 0.102 | 27.46 | 0.919 | 0.092 |
| **EvReflection (Ours)** | S+E | **28.63** | **0.947** | **0.018** | **28.57** | **0.946** | **0.050** | **30.07** | **0.927** | **0.048** | **29.09** | **0.940** | **0.036** |

*Table 2.* Quantitative evaluation on the real-captured EVR$^2$ benchmark. The dataset is stratified by glass thickness (T3: 3 mm, T5: 5 mm, T8: 8 mm). Methods marked with ∗ are evaluated using official pre-trained models due to unavailable training code. EvReflection demonstrates consistent superiority across all subsets, with increasing advantages on thicker glass.

| Methods | EVR$^2$-T3 | | | EVR$^2$-T5 | | | EVR$^2$-T8 | | | Average | | |
|---|---|---|---|---|---|---|---|---|---|---|---|---|
| | PSNR↑ | SSIM↑ | LPIPS↓ | PSNR↑ | SSIM↑ | LPIPS↓ | PSNR↑ | SSIM↑ | LPIPS↓ | PSNR↑ | SSIM↑ | LPIPS↓ |
| IBCLN (Li et al., 2020) | 21.76 | 0.802 | 0.127 | 20.23 | 0.789 | 0.138 | 19.79 | 0.755 | 0.142 | 20.59 | 0.782 | 0.136 |
| YTMT (Hu & Guo, 2021) | 21.85 | 0.808 | 0.114 | 20.17 | 0.791 | 0.127 | 19.71 | 0.759 | 0.131 | 20.58 | 0.786 | 0.124 |
| LANet∗ (Dong et al., 2021) | 22.91 | 0.813 | 0.108 | 22.04 | 0.801 | 0.127 | 21.33 | 0.768 | 0.119 | 22.10 | 0.794 | 0.118 |
| SOLD∗ (Liu et al., 2021) | 22.21 | 0.806 | 0.127 | 20.46 | 0.786 | 0.143 | 19.94 | 0.755 | 0.146 | 20.87 | 0.783 | 0.138 |
| DSRNet (Hu & Guo, 2023) | 25.50 | 0.873 | 0.085 | 24.50 | 0.856 | 0.099 | 23.07 | 0.808 | 0.102 | 24.35 | 0.845 | 0.095 |
| RRW (Zhu et al., 2024) | 26.17 | 0.882 | 0.115 | 24.94 | 0.869 | 0.125 | 24.03 | 0.825 | 0.126 | 25.05 | 0.859 | 0.122 |
| DSIT (Hu et al., 2024) | 27.04 | 0.904 | 0.068 | 25.20 | 0.889 | 0.081 | 24.45 | 0.845 | 0.083 | 25.65 | 0.880 | 0.077 |
| RDNet (Zhao et al., 2025) | 27.68 | 0.902 | 0.063 | 25.93 | 0.887 | 0.075 | 24.32 | 0.837 | 0.079 | 25.97 | 0.875 | 0.072 |
| DAI∗ (Hu et al., 2026) | 22.81 | 0.817 | 0.105 | 20.74 | 0.801 | 0.119 | 20.20 | 0.769 | 0.127 | 21.25 | 0.796 | 0.117 |
| **EvReflection (Ours)** | **28.88** | **0.918** | **0.055** | **27.04** | **0.906** | **0.067** | **25.81** | **0.865** | **0.068** | **27.25** | **0.896** | **0.063** |

total of 420 paired sequences (140 scenes × 3 thicknesses) for comprehensive evaluation. Instead of simple random splitting, we employed a performance-guided stratified sampling strategy to select the test set. By mapping the error distribution of a baseline model, we curated 11 representative evaluation scenes that cover the full difficulty spectrum, ranging from faint, translucent reflections to heavy, saturated interference. These samples are organized into three thickness-based subsets (EVR$^2$-T3, T5, T8) for a detailed analysis of robustness against different ghosting severities.

### 4.2. Implementation Details

**Training Configurations.** We implement our model using the PyTorch framework. The training is conducted on a cluster of 8 NVIDIA RTX 4090 GPUs. We set the batch size to 2 per GPU, resulting in a total batch size of 16. The network parameters are optimized using the Adam optimizer (Kingma & Ba, 2015) with standard hyperparameters ($\beta_1 = 0.9, \beta_2 = 0.999$) and no weight decay. The initial learning rate is set to $1 \times 10^{-4}$ and is modulated by a Cosine Annealing (Loshchilov & Hutter, 2017) strategy over 55 epochs. To enhance generalization, we apply stan-

dard data augmentations, including random cropping and horizontal flipping, during the training phase.

**Loss Functions.** We strictly adhere to the training objective of RDNet (Zhao et al., 2025) to ensure fair comparison. The total loss comprises a weighted combination of reconstruction loss (integrating MSE and gradient terms) and VGG-based perceptual loss (Johnson et al., 2016). This composite objective enforces structural fidelity in separated layers while effectively suppressing visual artifacts.

### 4.3. Comparison with State-of-the-arts

**Evaluation on Synthetic Dataset.** We benchmark EvReflection against 11 state-of-the-art approaches, encompassing both single-image and multi-image methods, as listed in Table 1. It is worth noting that our model is trained exclusively on a compact synthetic dataset consisting of 7,643 image pairs from PASCAL VOC, with corresponding event streams generated via our parallax-aware simulation. In stark contrast, most competing baselines benefit from significantly larger-scale training sets that often include real-world data, *e.g.*, Real (Zhang et al., 2018) and Nature (Li

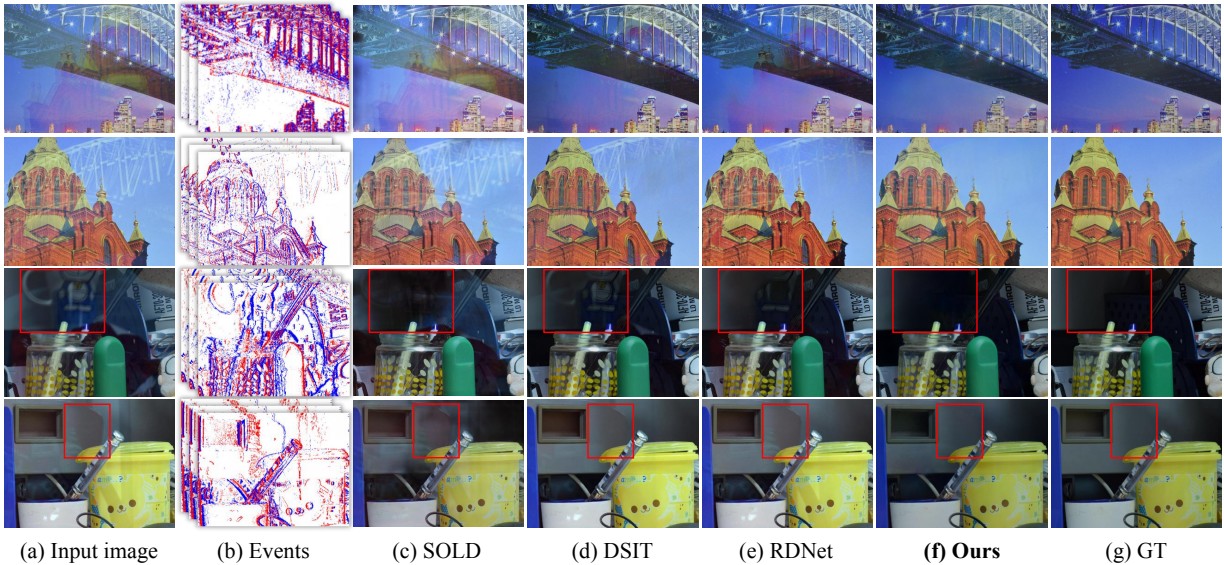

*Figure 6.* Qualitative comparison on the SIR$^2$ benchmark ([Wan et al., 2017](#)). Competing methods leave obvious residual artifacts in the zoomed regions (red boxes), while our method cleanly removes reflections and restores sharp details consistent with the ground truth.

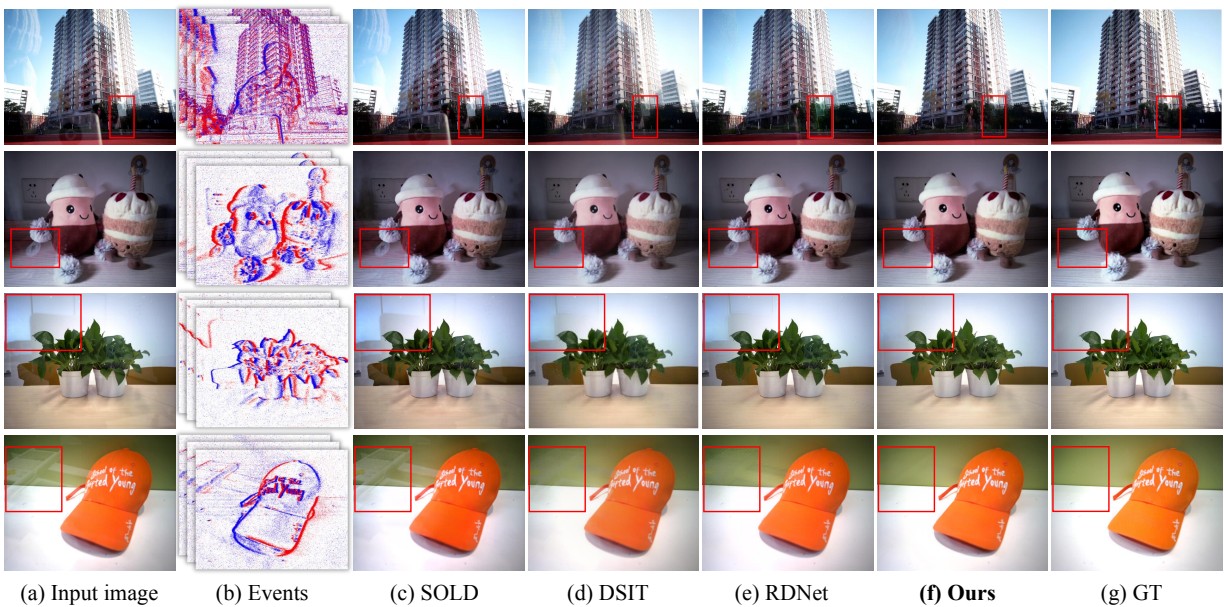

*Figure 7.* Qualitative comparison on the real-world EVR$^2$ benchmark. Competing methods fail to remove severe ghosting artifacts caused by glass thickness (red boxes), whereas our method cleanly eliminates these reflections and restores clear background details.

et al., 2020). Despite this apparent data disadvantage, our method demonstrates superior generalization, validating the high fidelity of our simulation pipeline.

As reported in Table 1, we evaluate performance on the SIR$^2$ benchmark using PSNR, SSIM, and LPIPS. Quantitatively, EvReflection outperforms all baselines by a significant margin across all subsets. Qualitatively, as depicted in Figure 6, our method overcomes the texture confusion inherent in single-image baselines by leveraging high-temporal-

resolution events to distinguish overlapping features. Furthermore, unlike multi-image approaches suffering from blurred edges due to registration errors, our alignment-free motion prior avoids such artifacts and preserves sharp structural details of the recovered transmission layer.

**Evaluation on Real-captured Dataset.** While our model trained solely on synthetic data demonstrates impressive generalization, real-world scenarios introduce complex factors such as sensor noise and varying refractive indices. To

*Table 3.* Ablation studies on $EVR^2$ investigating the event modality and key modules. **DC**: Dynamics Cues (*e.g.*, Optical Flow or MDD). Our full model (e) achieves the best performance, validating the effectiveness of each proposed component.

| Variant | Events | Dynamics Strategy | Fusion Strategy | PSNR↑ | SSIM↑ |
|---|---|---|---|---|---|
| (a) RGB Baseline | ✗ | – | – | 26.26 | 0.885 |
| (b) w/o DC | ✓ | – | PAR | 26.74 | 0.887 |
| (c) w/ Optical Flow | ✓ | Flow | PAR | 27.09 | **0.896** |
| (d) w/o PAR (Concat) | ✓ | MDD | Concat | 27.02 | 0.889 |
| **(e) Ours (Full)** | ✓ | **MDD** | **PAR** | **27.25** | **0.896** |

*Table 4.* Comparison with state-of-the-art networks from related event-based restoration tasks. All competing models are retrained on our $EVR^2$ dataset under identical settings. Our task-specific design outperforms generic event-based backbones.

| Models | Original Task | PSNR↑ | SSIM↑ | LPIPS↓ |
|---|---|---|---|---|
| EvLight | Low-Light Enhancement | 23.46 | 0.827 | 0.102 |
| DeblurSR | Super-Resolution | 23.61 | 0.827 | 0.102 |
| EFNet | Motion Deblurring | 24.24 | 0.855 | 0.153 |
| **EvReflection** | Reflection Removal | **27.25** | **0.896** | **0.063** |

bridge this domain gap, we employ a hybrid training strategy by combining the synthetic VOC dataset with 387 real-world samples from our $EVR^2$ benchmark dataset. This joint training allows the network to learn robust reflection physics from synthetic data while adapting to the specific noise patterns and ghosting effects of real-world event streams.

Table 2 reports the results on the difficulty-stratified $EVR^2$ test subsets. Our method establishes a new state-of-the-art, significantly surpassing conventional RGB-based approaches. Notably, we observe a consistent performance drop across all methods as glass thickness increases (from T3 to T8). This trend is physically attributed to the severe ghosting effect inherent to thick media: thicker glass induces larger spatial offsets between front and back surface reflections, creating complex double-reflection artifacts. Nevertheless, our method still significantly outperforms competitive baselines on the challenging 8mm subset, demonstrating the robustness of our event-guided strategy. Visual comparisons in Figure 7 further confirm this robustness. As highlighted by the red boxes, while RGB-only baselines struggle to remove such structural ghosts, our approach preserves the integrity of the transmission scene, showing superior restoration quality.

### 4.4. Ablation Studies

To validate the necessity of the event modality and evaluate the contribution of our key architectural components, we conduct a comprehensive ablation study on the $EVR^2$ dataset. The quantitative results are summarized in Table 3.

**Impact of Event Modality.** We first establish an RGB-only baseline (a) by removing the event branch. As shown in Table 3, this variant yields the lowest performance (26.26 dB). This performance gap confirms that relying solely on static intensity data is insufficient to resolve the inherent ambiguity between reflection and transmission layers. The event modality proves indispensable by providing complementary temporal cues to disambiguate the two layers.

**Effectiveness of Dynamics Cues.** We investigate different strategies for utilizing event signals. Model (b), which treats events merely as static texture cues without dynamics decoupling, fails to fully exploit the temporal precision of

the sensor. Model (c) introduces explicit optical flow as dynamics cues. While it achieves a high SSIM comparable to our method, its PSNR is lower because explicit flow estimation is often unstable in texture-less or repetitive regions. In contrast, our full model (e) employs MDD to learn high-dimensional implicit dynamics. This approach avoids the collapse into unreliable 2D vectors, achieving the best overall fidelity with a PSNR of 27.25 dB.

**Effectiveness of Fusion Strategy.** Finally, we compare fusion mechanisms. Replacing PAR with simple feature concatenation, as in model (d), results in a performance decline. This comparison demonstrates that a naive linear combination is insufficient for this task. Our PAR module uses the dynamic priors to spatially modulate the features, explicitly guiding the network to suppress reflection artifacts while preserving transmission details.

### 4.5. Comparison with Event-based Baselines

Since we are the first to introduce event cameras for reflection removal, there are no direct competitors. To validate the superiority of our specialized design, we compare EvReflection against state-of-the-art networks from related event-based tasks, including EvLight (Liang et al., 2024) for low-light enhancement, DeblurSR (Song et al., 2024) for super-resolution, and EFNet (Sun et al., 2022) for motion deblurring. All models are retrained on our $EVR^2$ dataset.

As reported in Table 4, these generic restoration networks perform suboptimally on the reflection removal task. The fundamental limitation is that these methods typically assume a single degradation model, *e.g.*, noise or blur, and aim to enhance the overall signal. They lack specific mechanisms to decouple two additive image layers, i.e., reflection and transmission. In contrast, our method explicitly models the differential motion between layers, thereby achieving significantly superior performance.

## 5. Conclusion

In this paper, we present EvReflection, the first event-driven framework designed to tackle the inherent ambiguity of single-image reflection removal. By exploiting the high-temporal-resolution micro-dynamics captured by event cam-

eras, our method effectively disentangles the mixed reflection and transmission layers. We introduce a specialized cross-modal architecture featuring a Micro-Dynamics Decoupler (MDD) and a Parallax-Attention Rectifier (PAR), which robustly separate layer-specific motions without relying on strict alignment. Furthermore, we contribute the $EVR^2$ benchmark dataset to bridge the data gap in this domain. Extensive experiments on both synthetic and real-world datasets demonstrate that our approach significantly outperforms state-of-the-art RGB-based methods and generic event-based networks, achieving gains of over 1.6 dB and 1.2 dB in PSNR on the $SIR^2$ and $EVR^2$ benchmarks, respectively. This work establishes a solid foundation for future research in event-guided image separation tasks.

**Limitation.** EvReflection has three limitations. First, it struggles in strictly static scenes (*e.g.*, tripod setups) where minimal motion yields no useful event signal. Second, extreme low-light conditions introduce severe sensor noise that degrades both modalities. Lastly, the large backbone incurs substantial computational overhead. In future work, we plan to address these by incorporating motion-free priors and developing lightweight alternatives.

## Acknowledgements

This work was in part supported by the National Natural Science Foundation of China under grants 62472399 and Open Fund of APKL of BIIP, IAI, Hefei Comprehensive National Science Center under grants 24YGXT003.

## Impact Statement

This paper presents work whose goal is to advance the field of Machine Learning. There are many potential societal consequences of our work, none of which we feel must be specifically highlighted here.

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

# A. Theoretical Analysis of RGB-based Limitations

## A.1. The Ill-posedness of Single-image Reflection Removal

The reflection superposition is classically modeled as:

$$\mathbf{I}(\mathbf{x}) = \mathbf{T}(\mathbf{x}) + \mathbf{R}(\mathbf{x}), \quad \forall \mathbf{x} \in \Omega \tag{7}$$

where $\mathbf{I}(\mathbf{x})$, $\mathbf{T}(\mathbf{x})$, and $\mathbf{R}(\mathbf{x})$ denote the observed intensity of the mixed, transmission, and reflection scenes, respectively, accumulated over the exposure time $\Delta t$ within the spatial domain $\Omega$, where $\mathbf{x} = (x, y)$ represents the pixel coordinates. Recovering $\mathbf{T}$ and $\mathbf{R}$ from a single observation $\mathbf{I}$ is a fundamentally ill-posed inverse problem, as there exist infinite pairs of $(\mathbf{T}, \mathbf{R})$ satisfying Equation (7).

## A.2. Limitations of Frame-based Motion Constraints

To resolve this ambiguity, multi-frame approaches introduce temporal dynamics. Given that frame-based cameras integrate intensity over the exposure time, the reflection superposition model is formulated as:

$$\begin{aligned}
\mathbf{I}(\mathbf{x}) &= \int_0^{\Delta t} \mathcal{I}(\mathbf{x} - \mathbf{u}\tau) \, d\tau \\
&= \int_0^{\Delta t} \left( \mathcal{T}(\mathbf{x} - \mathbf{u}_T \tau) + \mathcal{R}(\mathbf{x} - \mathbf{u}_R \tau) \right) d\tau,
\end{aligned} \tag{8}$$

where $\Delta t$ represents the camera exposure time. $\mathcal{I}$, $\mathcal{T}$, and $\mathcal{R}$ refer to the latent instantaneous radiance of the mixed, transmission, and reflection scenes, respectively. The vectors $\mathbf{u}_T$ and $\mathbf{u}_R$ denote the corresponding velocity fields during exposure, and $\tau$ serves as the temporal integration variable. Mathematically, this integration is equivalent to a spatial convolution:

$$\mathbf{I}(\mathbf{x}) = \mathcal{T}(\mathbf{x}) * h_T(\mathbf{x}) + \mathcal{R}(\mathbf{x}) * h_R(\mathbf{x}), \tag{9}$$

where $h_T$ and $h_R$ are normalized boxcar kernels determined by the motion magnitude. Specifically, for a layer $k \in \{T, R\}$, the kernel profile along its motion direction $s$ is defined as:

$$h_k(s) = \begin{cases} \frac{1}{\|\mathbf{u}_k\|}, & \text{if } 0 \leq s \leq L_k \\ 0, & \text{otherwise}, \end{cases} \tag{10}$$

where $L_k = \|\mathbf{u}_k\|\Delta t$ denotes the blur length. Applying the convolution theorem transforms the spatial integration in Equation (9) into frequency-domain multiplications. Since the Fourier transform of a boxcar kernel is a Sinc function, the observed spectrum is derived as:

$$\hat{\mathbf{I}}(\omega) = \text{sinc}\left(\frac{\omega \cdot L_T}{2}\right) \hat{\mathcal{T}}(\omega) + \text{sinc}\left(\frac{\omega \cdot L_R}{2}\right) \hat{\mathcal{R}}(\omega), \tag{11}$$

where $\hat{\mathbf{I}}$, $\hat{\mathcal{T}}$, and $\hat{\mathcal{R}}$ are the frequency spectra of the observed and latent images, respectively, and $\omega$ denotes the spatial frequency. Equation (11) reveals a fundamental **motion-bandwidth paradox** that limits frame-based reflection removal:

- When the relative motion is minimal ($L_k \to 0$), the system lacks sufficient motion parallax to geometrically distinguish the transmission from the reflection, causing the problem to degenerate into an ill-posed single-image separation.

- Conversely, increasing velocity to induce parallax leads to a larger blur length $L_k$. Since the main lobe width of the Sinc filter is proportional to $2\pi/L_k$, a larger $L_k$ narrows the passband. This acts as an aggressive low-pass filter, irreversibly attenuating the high-frequency components essential for edge-based separation.

## A.3. Theoretical Derivation of Ill-posedness in Mixed-image Optical Flow

In this section, we provide a rigorous derivation demonstrating why standard optical flow estimation fails when applied to reflection-contaminated images.

**Derivation for Superposition Model.** The brightness constancy assumption, which underpins most optical flow algorithms, states that the intensity of a pixel remains constant over time. For a generic image $I(\mathbf{x}, t)$, this is expressed as:

$$\frac{dI}{dt} = \frac{\partial I}{\partial t} + \nabla I \cdot \mathbf{v} = 0, \tag{12}$$

where $\mathbf{v}$ is the estimated optical flow vector.

In the reflection removal task, the observed image is a superposition of two layers: $I(\mathbf{x}, t) = T(\mathbf{x}, t) + R(\mathbf{x}, t)$. Substituting this into the total derivative yields:

$$\frac{\partial(T + R)}{\partial t} + \nabla(T + R) \cdot \mathbf{v} = 0. \tag{13}$$

Let $\mathbf{u}_T$ and $\mathbf{u}_R$ denote the ground-truth velocity fields for the transmission and reflection layers, respectively. The temporal derivatives of the individual layers satisfy their own continuity equations:

$$\frac{\partial T}{\partial t} = -\nabla T \cdot \mathbf{u}_T, \quad \frac{\partial R}{\partial t} = -\nabla R \cdot \mathbf{u}_R. \tag{14}$$

Substituting these relationships back into the mixed-image constraint equation:

$$\begin{aligned}
(-\nabla T \cdot \mathbf{u}_T - \nabla R \cdot \mathbf{u}_R) + (\nabla T + \nabla R) \cdot \mathbf{v} &= 0 \\
\nabla T \cdot (\mathbf{v} - \mathbf{u}_T) + \nabla R \cdot (\mathbf{v} - \mathbf{u}_R) &= 0.
\end{aligned} \tag{15}$$

The optical flow objective function $\mathcal{L}(\mathbf{v})$ minimizes the squared magnitude of this residual:

$$\mathcal{L}(\mathbf{v}) = |\nabla T \cdot (\mathbf{v} - \mathbf{u}_T) + \nabla R \cdot (\mathbf{v} - \mathbf{u}_R)|^2. \tag{16}$$

**Analysis of the Conflict.** To achieve a zero loss ($\mathcal{L}(\mathbf{v}) = 0$), the estimated flow $\mathbf{v}$ must satisfy both terms simultaneously. This implies:

$$\mathbf{v} = \mathbf{u}_T \quad \text{and} \quad \mathbf{v} = \mathbf{u}_R. \tag{17}$$

However, a fundamental premise of motion-based reflection removal is the existence of motion parallax, meaning $\mathbf{u}_T(\mathbf{x}) \neq \mathbf{u}_R(\mathbf{x})$ for the majority of pixels $\mathbf{x}$.

Mathematically, the optimizer seeks a solution $\mathbf{v}^*$ that minimizes the weighted sum of projections. The resulting $\mathbf{v}^*$ typically represents a weighted average of $\mathbf{u}_T$ and $\mathbf{u}_R$, heavily biased towards the layer with stronger gradients. Consequently:

- If $\|\nabla T\| \gg \|\nabla R\|$: $\mathbf{v}^* \approx \mathbf{u}_T$, causing the reflection layer to be misaligned.

- If $\|\nabla R\| \gg \|\nabla T\|$: $\mathbf{v}^* \approx \mathbf{u}_R$, causing the transmission layer to be distorted.

- If $\|\nabla T\| \approx \|\nabla R\|$: $\mathbf{v}^*$ aligns with neither, leading to tearing artifacts.

This derivation theoretically confirms that estimating a single rigid or non-rigid flow field from mixed observations is structurally incapable of capturing the distinct dynamics of the two layers.

## B. Theoretical Analysis of Event-based Geometric Constraints

### B.1. Problem Formulation and Analysis

In contrast to frame-based cameras, event cameras operate asynchronously, triggering discrete events $e_k = (\mathbf{x}_k, t_k, p_k)$ at timestamp $t_k$ whenever the logarithmic intensity change at pixel $\mathbf{x}_k$ exceeds a contrast threshold $C$:

$$\Delta \ln \mathcal{I}(\mathbf{x}, t) = \ln \mathcal{I}(\mathbf{x}, t_k) - \ln \mathcal{I}(\mathbf{x}, t_{k-1}) = p_k C, \tag{18}$$

where $\mathcal{I}$ refers to the latent instantaneous radiance of the mixed scene. For a microsecond-level temporal window, the discrete event stream can be approximated as a continuous spatiotemporal signal $\mathcal{E}(\mathbf{x}, t)$, representing the temporal derivative of the logarithmic intensity:

$$\mathcal{E}(\mathbf{x}, t) = \frac{\partial \ln \mathcal{I}(\mathbf{x}, t)}{\partial t} = \frac{1}{\mathcal{I}(\mathbf{x}, t)} \frac{\partial \mathcal{I}(\mathbf{x}, t)}{\partial t} = \frac{1}{\mathcal{T} + \mathcal{R}} \left( \frac{\partial \mathcal{T}}{\partial t} + \frac{\partial \mathcal{R}}{\partial t} \right). \tag{19}$$

Assuming the brightness constancy constraint holds locally for each layer, the temporal derivative relates to the spatial gradient via the OFCE:

$$\frac{\partial \mathcal{T}}{\partial t} = -\langle \mathbf{u}_T, \nabla \mathcal{T} \rangle, \quad \frac{\partial \mathcal{R}}{\partial t} = -\langle \mathbf{u}_R, \nabla \mathcal{R} \rangle. \tag{20}$$

Here, $\langle \cdot, \cdot \rangle$ denotes the inner product operation, and $\nabla \mathcal{T}, \nabla \mathcal{R} \in \mathbb{R}^2$ denote the spatial gradients ($[\frac{\partial}{\partial x}, \frac{\partial}{\partial y}]^\top$) of the transmission and reflection layers, respectively. Combining Equation (19) and Equation (20) yields the **Event-Gradient Constraint** partial differential equation:

$$\mathcal{I} \mathcal{E} = - \left( \langle \mathbf{u}_T, \nabla \mathcal{T} \rangle + \langle \mathbf{u}_R, \nabla \mathcal{R} \rangle \right). \tag{21}$$

Physically, this implies that the observed event intensity corresponds to the negative weighted sum of the spatial gradients projected along their respective motion directions.

We aggregate observations within a local spatiotemporal neighborhood $\mathcal{N}(\mathbf{x}, t)$. We enforce a collinear motion constraint $\mathbf{u}_T = v_T \mathbf{d}$ and $\mathbf{u}_R = v_R \mathbf{d}$, where $v_T$ and $v_R$ denote the scalar motion magnitudes and $\mathbf{d}$ represents the shared unit direction vector. Assuming that the spatial gradients $\nabla \mathcal{T}$ and $\nabla \mathcal{R}$ remain constant within a micro-temporal window $[t_1, t_N]$, we construct a linear system $\mathbf{Ax} = \mathbf{b}$:

$$\underbrace{\begin{bmatrix} (v_T)_{t_1} & (v_R)_{t_1} \\ (v_T)_{t_2} & (v_R)_{t_2} \\ \vdots & \vdots \end{bmatrix}}_{\mathbf{A} \in \mathbb{R}^{N \times 2}} \underbrace{\begin{bmatrix} \langle \mathbf{d}, \nabla \mathcal{T} \rangle \\ \langle \mathbf{d}, \nabla \mathcal{R} \rangle \end{bmatrix}}_{\mathbf{x} \in \mathbb{R}^2} = - \underbrace{\begin{bmatrix} \mathcal{I}_{t_1} \mathcal{E}_{t_1} \\ \mathcal{I}_{t_2} \mathcal{E}_{t_2} \\ \vdots \end{bmatrix}}_{\mathbf{b} \in \mathbb{R}^N}. \tag{22}$$

Defining $x_T^\parallel = \langle \mathbf{d}, \nabla \mathcal{T} \rangle$ and $x_R^\parallel = \langle \mathbf{d}, \nabla \mathcal{R} \rangle$, we aim to recover the optimal latent gradients $\mathbf{x} = [x_T^\parallel, x_R^\parallel]^\top$ by minimizing the observation residuals. This is formulated as the following convex optimization problem:

$$\min_{\mathbf{x}} \mathcal{L}(\mathbf{x}) = \frac{1}{2} \|\mathbf{Ax} - \mathbf{b}\|^2. \tag{23}$$

To ensure that the optimization problem formulated in Equation (23) is strictly convex and possesses a unique global solution, the Hessian matrix of the objective function, denoted as $\mathbf{H} = \mathbf{A}^\top \mathbf{A}$, must be positive definite. We provide the rigorous proof in the following subsection.

### B.2. Proof of Positive Definiteness and Unique Solution

Recall the structure of the Hessian matrix derived from the least-squares formulation $\mathbf{H} = \mathbf{A}^\top \mathbf{A}$:

$$\mathbf{H} = \begin{bmatrix} \langle \mathbf{S}_T, \mathbf{S}_T \rangle & \langle \mathbf{S}_T, \mathbf{S}_R \rangle \\ \langle \mathbf{S}_R, \mathbf{S}_T \rangle & \langle \mathbf{S}_R, \mathbf{S}_R \rangle \end{bmatrix} = \begin{bmatrix} \|\mathbf{S}_T\|^2 & \mathbf{S}_T^\top \mathbf{S}_R \\ \mathbf{S}_R^\top \mathbf{S}_T & \|\mathbf{S}_R\|^2 \end{bmatrix}, \tag{24}$$

where $\mathbf{S}_T, \mathbf{S}_R \in \mathbb{R}^N$ denote the aggregated velocity magnitude vectors over the observation window $\mathcal{N}$, defined as:

$$\mathbf{S}_T = [(v_T)_{t_1}, (v_T)_{t_2}, \ldots, (v_T)_{t_N}]^\top, \quad \mathbf{S}_R = [(v_R)_{t_1}, (v_R)_{t_2}, \ldots, (v_R)_{t_N}]^\top. \tag{25}$$

**Positive Definiteness Condition.** According to Sylvester's Criterion, a symmetric matrix is positive definite if and only if all its leading principal minors are positive. For the $2 \times 2$ Hessian matrix $\mathbf{H}$, the conditions are:

- First leading principal minor: $H_{11} > 0$.

- Second leading principal minor (Determinant): $\det(\mathbf{H}) > 0$.

**Verification of First Minor.** The first element corresponds to the squared Euclidean norm of the transmission velocity vector:

$$H_{11} = \|\mathbf{S}_T\|^2 = \sum_{i=1}^{N} (v_T)_{t_i}^2. \tag{26}$$

Since the camera motion (even slight hand tremor) is non-zero during the event accumulation interval, the velocity magnitude vector is non-vanishing ($\mathbf{S}_T \neq \mathbf{0}$). Thus, strict positivity holds:

$$\|\mathbf{S}_T\|^2 > 0 \implies H_{11} > 0. \tag{27}$$

**Verification of Determinant.** The determinant of $\mathbf{H}$ is calculated as:

$$\det(\mathbf{H}) = H_{11}H_{22} - H_{12}H_{21} = \|\mathbf{S}_T\|^2 \|\mathbf{S}_R\|^2 - \langle \mathbf{S}_T, \mathbf{S}_R \rangle^2. \tag{28}$$

The Cauchy-Schwarz inequality implies that for any two vectors $\mathbf{u}, \mathbf{v}$ in an inner product space:

$$\langle \mathbf{u}, \mathbf{v} \rangle^2 \leq \|\mathbf{u}\|^2 \|\mathbf{v}\|^2. \tag{29}$$

Here, equality is achieved if and only if $\mathbf{u}$ and $\mathbf{v}$ are linearly dependent (i.e., $\mathbf{u} = k\mathbf{v}$). Applying this to our context:

$$\det(\mathbf{H}) \geq 0. \tag{30}$$

To prove strictly positive definiteness ($\det(\mathbf{H}) > 0$), we must demonstrate that $\mathbf{S}_T$ and $\mathbf{S}_R$ are linearly independent. The vectors $\mathbf{S}_T$ and $\mathbf{S}_R$ represent the temporal evolution of optical flow magnitudes. Based on the motion parallax model under general camera motion (including rotation or translation), the optical flow $\mathbf{u}(\mathbf{x}, t)$ is inversely proportional to the depth $Z(t)$:

$$\mathbf{u}_T(\mathbf{x}, t) \propto \frac{1}{Z_T(t)}, \quad \mathbf{u}_R(\mathbf{x}, t) \propto \frac{1}{Z_R(t)}. \tag{31}$$

Given that the transmission scene and the virtual reflection image usually reside at distinct depths ($Z_T \neq Z_R$) and the camera trajectory includes non-trivial dynamics, the induced velocity fields differ in magnitude scaling over time. Consequently, the aggregated velocity vectors satisfy:

$$\mathbf{S}_T \neq k \cdot \mathbf{S}_R, \quad \forall k \in \mathbb{R}. \tag{32}$$

This implies strict linear independence. Therefore, the equality condition of the Cauchy-Schwarz inequality cannot be met, leading to:

$$\|\mathbf{S}_T\|^2 \|\mathbf{S}_R\|^2 - \langle \mathbf{S}_T, \mathbf{S}_R \rangle^2 > 0 \implies \det(\mathbf{H}) > 0. \tag{33}$$

### B.3. Conclusion

Since both leading principal minors are strictly positive:

$$H_{11} > 0 \quad \text{and} \quad \det(\mathbf{H}) > 0, \tag{34}$$

we conclude that the Hessian matrix $\mathbf{H}$ is strictly positive definite ($\mathbf{H} \succ 0$). This confirms that the optimization landscape is strongly convex, guaranteeing the existence of a unique global minimizer for the projected gradients $x_T^{\parallel}$ and $x_R^{\parallel}$.

Crucially, this result signifies a major advantage over single-image RGB methods. While recovering the transmission and reflection layers from a single image is inherently ill-posed due to the infinite valid decompositions, the introduction of event signals imposes a strict physical constraint governed by the non-zero motion parallax. The unique determination of the projected gradients significantly constrains the feasible solution space, effectively transforming the under-determined problem into a well-posed one within the gradient domain, thereby reducing the ambiguity in the final restoration.

*Table C.1.* Supplementary ablation studies on $EVR^2$ subsets. We report PSNR, SSIM, and LPIPS to demonstrate the effectiveness of our MDD and PAR modules across different scenarios. The best and second-best scores are marked in **red** and blue, respectively.

| Variant | Events | Dynamics Strategy | Fusion Strategy | $EVR^2$-T3 | | | $EVR^2$-T5 | | | $EVR^2$-T8 | | |
|---|---|---|---|---|---|---|---|---|---|---|---|---|
| | | | | PSNR↑ | SSIM↑ | LPIPS↓ | PSNR↑ | SSIM↑ | LPIPS↓ | PSNR↑ | SSIM↑ | LPIPS↓ |
| (a) w/o Event | ✗ | – | – | 28.06 | 0.911 | 0.060 | 25.98 | 0.895 | 0.073 | 24.63 | 0.846 | 0.074 |
| (b) w/o DC | ✓ | – | PAR | 28.50 | 0.911 | 0.058 | 26.62 | 0.899 | 0.069 | 25.00 | 0.849 | 0.072 |
| (c) w/ Optical Flow | ✓ | Flow | PAR | 28.84 | **0.920** | **0.055** | 26.76 | **0.907** | 0.068 | 25.48 | 0.858 | 0.069 |
| (d) w/o PAR (Concat) | ✓ | MDD | Concat | 28.74 | 0.918 | 0.057 | 26.68 | **0.907** | 0.069 | 25.51 | 0.862 | 0.069 |
| (e) Ours (Full) | ✓ | **MDD** | **PAR** | **28.88** | 0.918 | **0.055** | **27.04** | 0.906 | **0.067** | **25.81** | **0.865** | **0.063** |

*Table C.2.* Detailed comparison with other event-based restoration baselines on $EVR^2$ subsets. This breakdown complements the averaged results in the main paper by evaluating cross-task generalization on different glass thicknesses.

| Models | Original Task | $EVR^2$-T3 | | | $EVR^2$-T5 | | | $EVR^2$-T8 | | |
|---|---|---|---|---|---|---|---|---|---|---|
| | | PSNR↑ | SSIM↑ | LPIPS↓ | PSNR↑ | SSIM↑ | LPIPS↓ | PSNR↑ | SSIM↑ | LPIPS↓ |
| EvLight (Liang et al., 2024) | Low-Light Enhancement | 24.44 | 0.852 | 0.098 | 23.44 | 0.834 | 0.103 | 22.51 | 0.796 | 0.105 |
| DeblurSR (Song et al., 2024) | Super-Resolution | 24.41 | 0.851 | 0.098 | 23.71 | 0.834 | 0.102 | 22.69 | 0.795 | 0.105 |
| EFNet (Sun et al., 2022) | Motion Deblurring | 25.09 | 0.876 | 0.147 | 24.12 | 0.862 | 0.155 | 23.51 | 0.826 | 0.156 |
| **EvReflection (Ours)** | Reflection Removal | **28.88** | **0.918** | **0.055** | **27.04** | **0.906** | **0.067** | **25.81** | **0.865** | **0.063** |

# C. Full Quantitative Results on $EVR^2$ Subsets

In this section, we present detailed quantitative evaluations on three subsets of the real-world $EVR^2$ dataset: $EVR^2$-T3, $EVR^2$-T5, and $EVR^2$-T8, corresponding to glass thicknesses of 3mm, 5mm, and 8mm, respectively. We perform two sets of experiments: comprehensive ablation studies to validate our architectural components (shown in Table C.1) and comparative experiments with cross-domain event-based baselines (shown in Table C.2). Note that a larger glass thickness implies a larger spatial offset between the reflection and transmission layers, making the separation more challenging.

## C.1. Ablation Analysis on Subsets

Table C.1 highlights the necessity of the event modality and the effectiveness of our MDD and PAR modules, particularly in challenging scenarios with thicker glass. Comparing rows (a) and (e), the performance gap widens as the glass thickness increases. On $EVR^2$-T3, introducing events improves PSNR by +0.82 dB (28.06 dB vs. 28.88 dB), while on the most difficult subset, $EVR^2$-T8, the gain increases to +1.18 dB (24.63 dB vs. 25.81 dB). This indicates that event signals provide critical high-frequency cues for handling severe ghosting artifacts that RGB-only methods fail to resolve.

**Effectiveness of MDD and PAR.** Comparing row (d) (Concat) with row (e) (Ours), our PAR module consistently outperforms the direct concatenation strategy. For instance, on $EVR^2$-T5, our full model achieves a PSNR of 27.04 dB, surpassing the 'w/o PAR' baseline by +0.36 dB. This suggests that the attention-based rectification, guided by the disentangled motion priors from MDD, effectively suppresses residual reflection features that simple fusion cannot eliminate.

## C.2. Comparison with Cross-Domain Baselines on Subsets

Table C.2 details the performance comparison against state-of-the-art event-based networks from other domains. Consistent with the observations in the ablation study, all methods exhibit a performance decline as the glass thickness increases from 3mm (T3) to 8mm (T8), confirming that larger spatial offsets pose a greater challenge for feature alignment and fusion.

Despite this general trend, our method demonstrates superior robustness compared to the adapted baselines. On the $EVR^2$-T3 subset, where the spatial offset is relatively small and the ghosting artifacts resemble typical motion blur, the motion deblurring method EFNet (Sun et al., 2022) achieves a reasonable PSNR of 25.09 dB. However, our EvReflection still leads the nearest competitor by a substantial margin of +3.79 dB (28.88 dB vs. 25.09 dB), indicating that our specialized reflection modeling is highly beneficial even in mild scenarios.

The robustness of our approach becomes even more critical on the challenging $EVR^2$-T8 subset. While generic restoration networks suffer from severe performance degradation due to their inability to distinguish the decoupled motions of reflection

and transmission (*e.g.*, EvLight (Liang et al., 2024) drops to 22.51 dB), our method maintains a reliable high-fidelity performance of 25.81 dB. This consistent advantage verifies that our task-specific design, specifically the proposed decoupling mechanism, is essential for handling the complex, non-linear superimposition found in real-world thick glass reflections, where simple domain adaptation strategies often fail.

## D. Event Representation

An event camera generates an asynchronous stream of events triggered by logarithmic brightness intensity changes. This stream can be mathematically denoted as $\mathcal{E} = \{e_k\}_{k=1}^K$, where $K$ is the total number of events within a given time window. Each event $e_k$ is a tuple $(x_k, y_k, t_k, p_k)$, representing the spatial coordinates, timestamp, and polarity $p_k \in \{+1, -1\}$, respectively. For notational brevity, we denote the spatial coordinates as a vector $\mathbf{x}_k = (x_k, y_k)$. However, since raw event streams are sparse and asynchronous, they are incompatible with standard CNNs.

To bridge this gap while preserving high-fidelity temporal information, we transform the event stream into a spatio-temporal voxel grid $\mathcal{V} \in \mathbb{R}^{B \times H \times W}$. Specifically, we discretize the time domain $[t_0, t_K]$ into $B$ temporal bins, and the voxel grid is constructed as:

$$\mathcal{V}(i) = \sum_{e_k \in \mathcal{E}} p_k \max\left(0, 1 - \left|i - \frac{t_k - t_0}{t_K - t_0}(B-1)\right|\right), \tag{35}$$

where $i \in \{0, \cdots, B-1\}$ denotes the index of the temporal bin. Unlike previous approaches (Zhu et al., 2019) that typically set $B = 5$, we increase the temporal resolution to $B = 20$. This ensures that the subtle motion parallax inherently present in the reflection and transmission layers is preserved, providing distinct dynamic cues for the subsequent decoupling. Finally, to suppress noise from hot pixels and outliers, we adopt a robust normalization strategy (Zhu et al., 2021):

$$\hat{\mathcal{V}} = \frac{\min(\mathcal{V}, \eta)}{\eta}, \tag{36}$$

where $\eta$ is the 98th percentile of the non-zero values in $\mathcal{V}$. The normalized tensor $\hat{\mathcal{V}}$ serves as the input to our network.

## E. More Qualitative Results

In this section, to further validate the effectiveness of EvReflection in handling diverse reflection scenarios, we provide extensive qualitative comparisons covering both synthetic and real-world distributions. The results demonstrate our model's superiority in removing complex reflections while preserving the integrity of the transmission layer.

**Results on Synthetic SIR$^2$ Dataset.** First, we evaluate the generalization capability of our model. We utilize the model trained on the synthetic VOC dataset and test it directly on the three subsets of the SIR$^2$ dataset: Objects, Postcard, and Wild. The visual results are presented in Figure E.1, Figure E.2, and Figure E.3, respectively.

- As shown in Figure E.1 (Objects), our method effectively eliminates strong specular highlights on solid objects, whereas baseline methods often leave distinct residual artifacts.

- In the texture-rich scenarios of Figure E.2 (Postcard), our method successfully distinguishes the high-frequency details of the transmission from the reflection interference, avoiding the over-smoothing problem common in other approaches.

- For the uncontrolled outdoor scenes in Figure E.3 (Wild), our model exhibits robust generalization, accurately recovering background structures even under complex lighting conditions.

**Results on Real-World EVR$^2$ Dataset.** Second, to assess the performance in real-world applications, we present results from the model trained and tested on our EVR$^2$ dataset. As illustrated in Figure E.4, real-world reflections often manifest as complex double-layer ghosting artifacts due to glass thickness. While state-of-the-art RGB-based methods rely solely on static spatial appearance priors (*e.g.*, ghosting cues or intensity differences), they often fail to distinguish strong reflections from the background, resulting in significant residual artifacts or over-smoothed transmission. In contrast, by incorporating the event modality, our EvReflection leverages the inherent high temporal resolution to capture the distinct independent dynamics of the reflection and transmission layers. This allows for precise disentanglement even in the presence of severe ghosting, yielding visually sharper and cleaner restoration results.

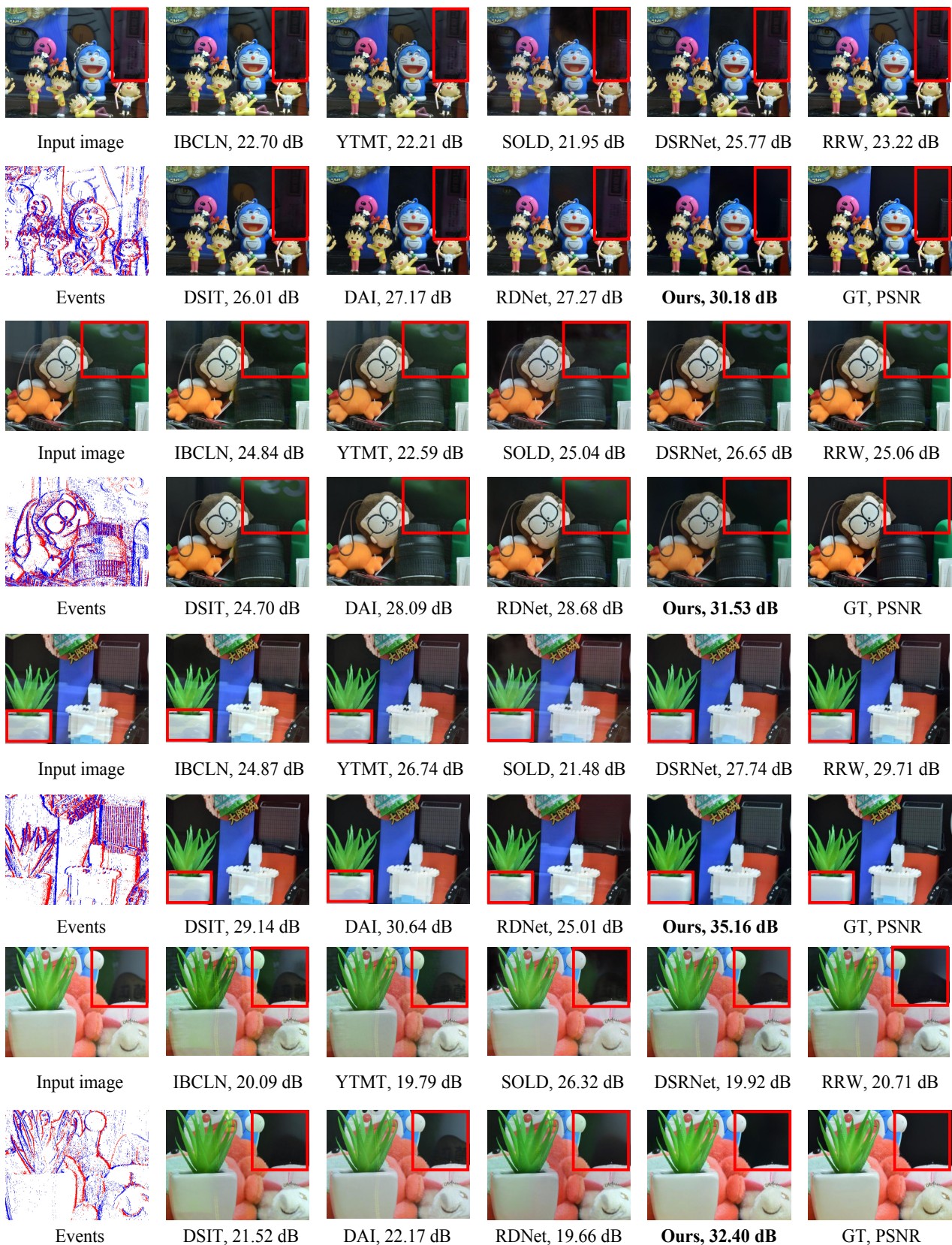

*Figure E.1.* Qualitative comparisons on the Objects subset of the synthetic SIR$^2$ dataset (Wan et al., 2017). **Zoomed in for best view.**

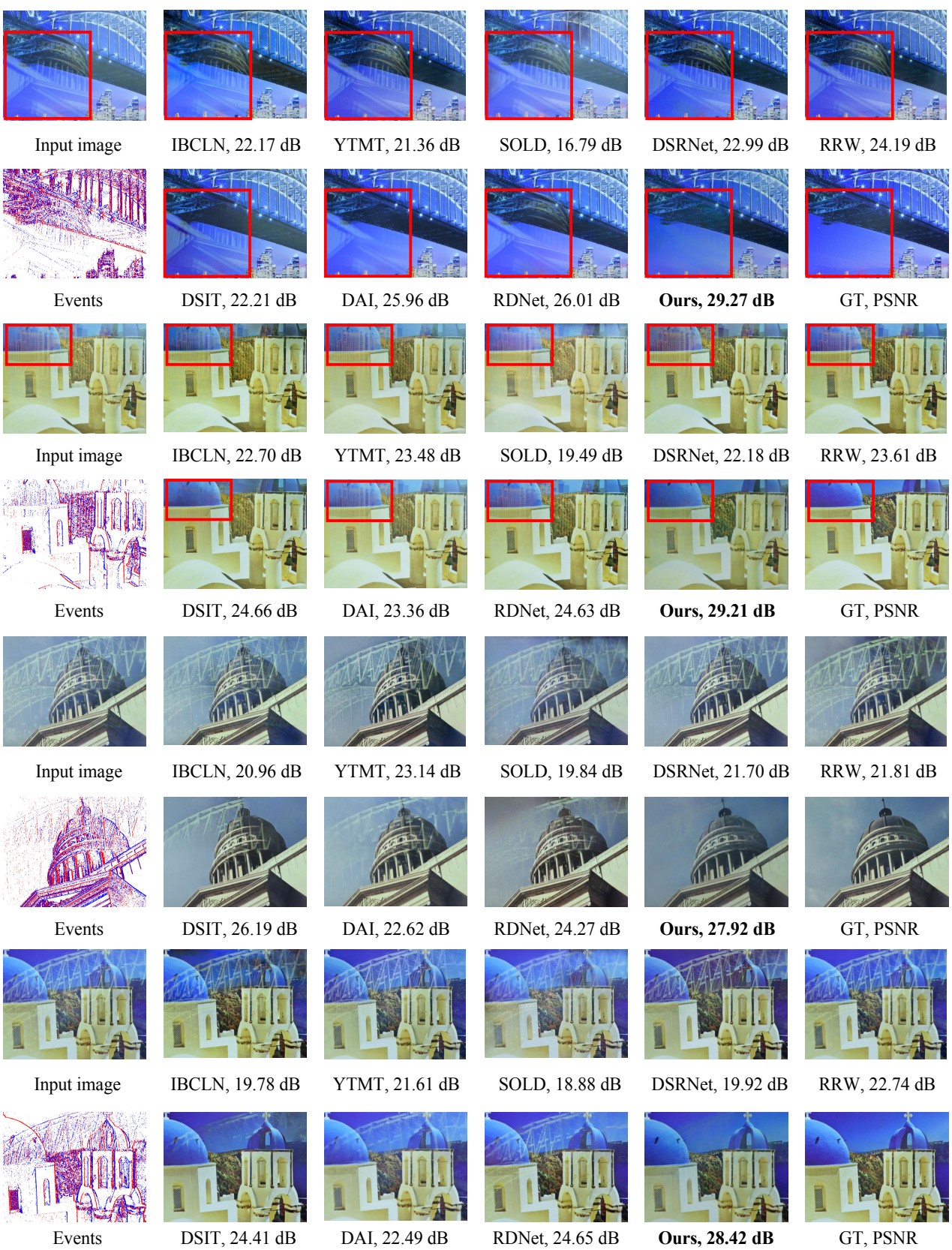

*Figure E.2.* Qualitative comparisons on the Postcard subset of the synthetic SIR$^2$ dataset (Wan et al., 2017). **Zoomed in for best view.**

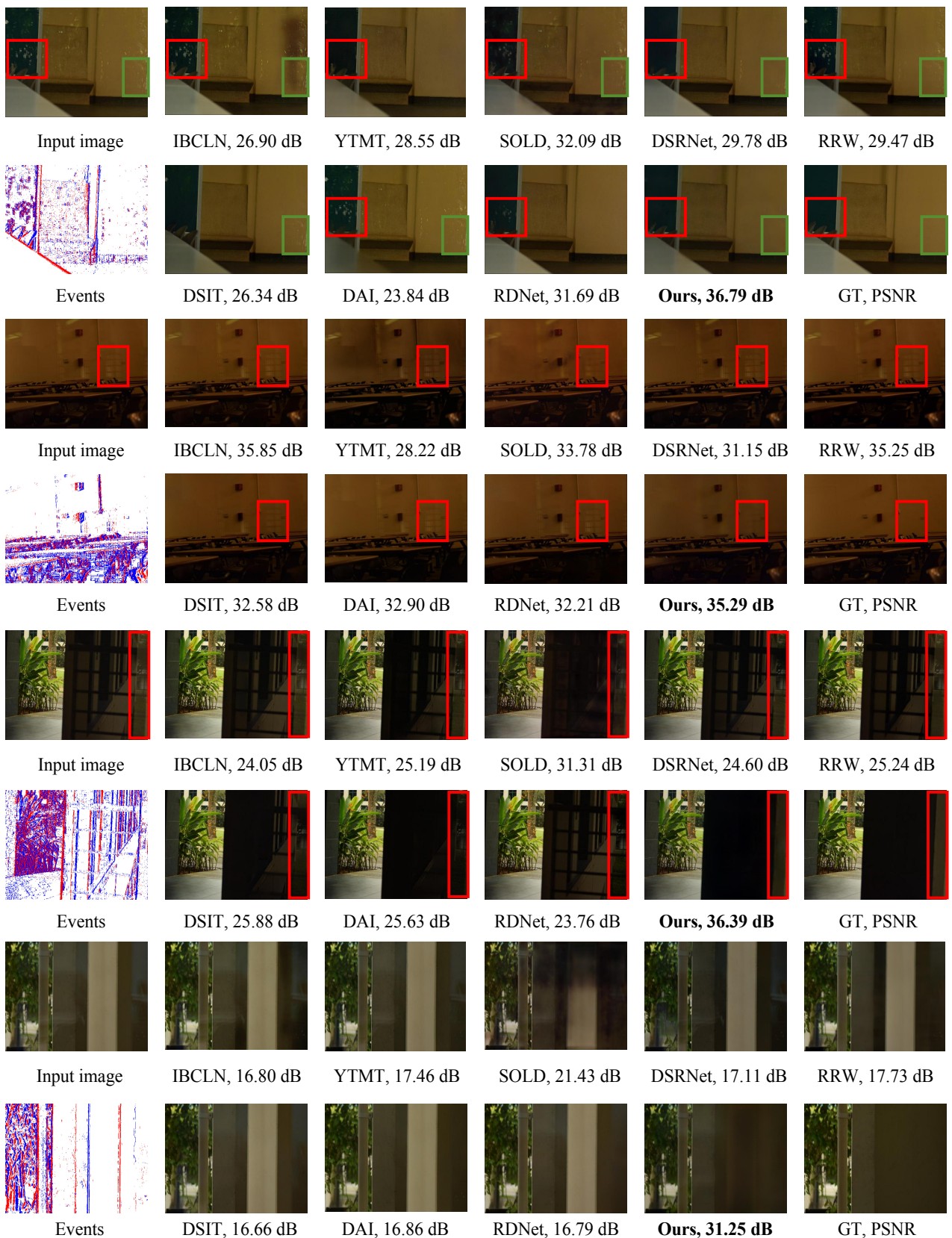

*Figure E.3.* Qualitative comparisons on the Wild subset of the synthetic SIR$^2$ dataset (Wan et al., 2017). **Zoomed in for best view.**

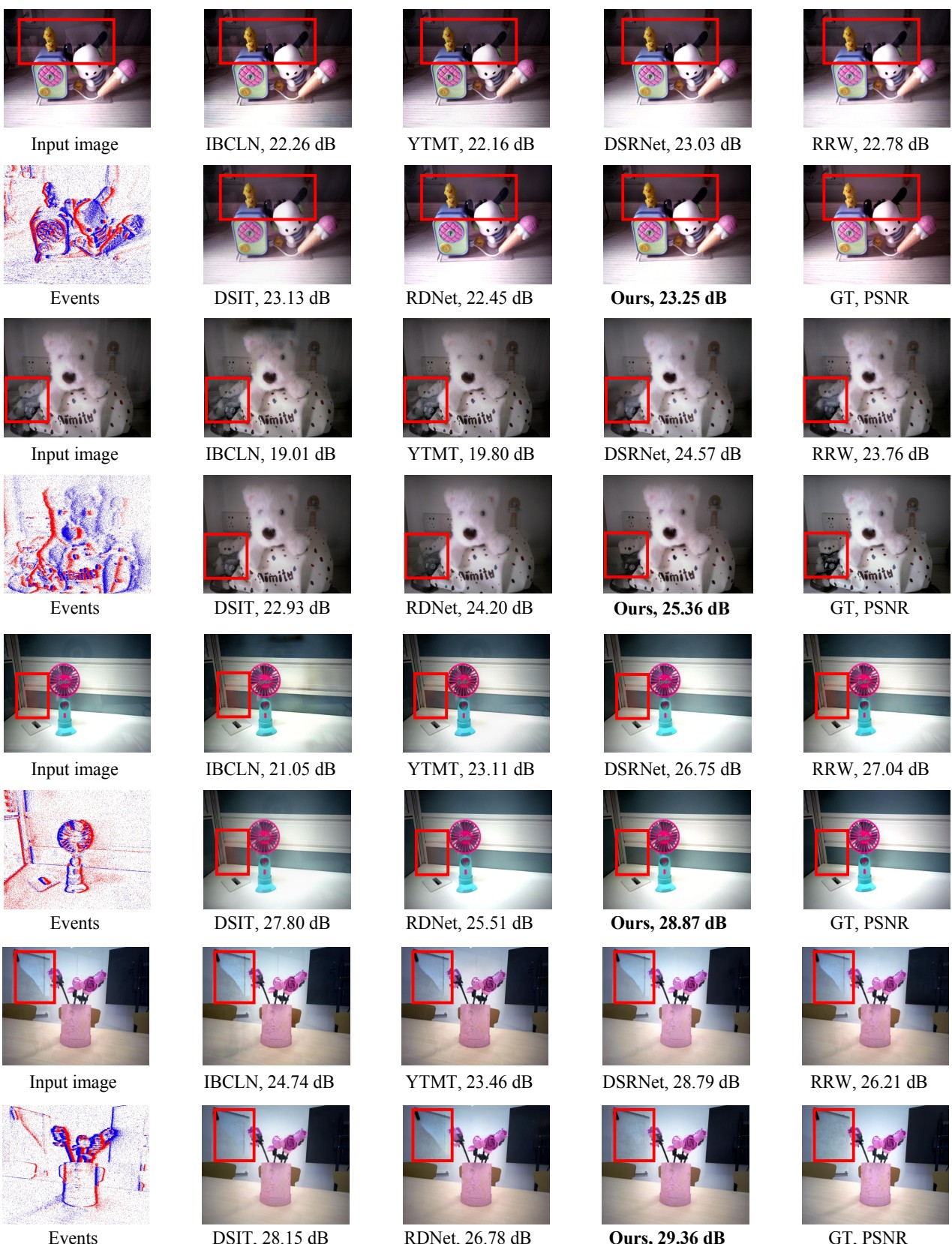

*Figure E.4.* Qualitative comparisons on our real-captured EVR² dataset. **Zoomed in for best view.**

# F. Model Details

In this section, we provide the specific architectural specifications required to reproduce our EvReflection network. The system adopts a hybrid design, leveraging a pre-trained FocalNet (Yang et al., 2022) as the image encoder, a RAFT-based motion extractor, and a specialized attention-based fusion module.

*Table F.1.* Detailed architectural specifications of EvReflection. The channel configurations for the Context Encoder refer to the dimensions after the adapter projection. $M$ indicates the number of iterations for the recurrent unit.

| Configuration | Context Encoder | MDD | PAR | Decoder |
|---|---|---|---|---|
| Base Architecture | FocalNet-L | RAFT-Variant | Dual-Channel Block | NAFBlock |
| Input Resolution | $H \times W$ | $H/8$ | $H/16$ | $H \times W$ |
| Depth / Iterations | [2, 2, 18, 2] | $M = 12$ | 1 | [1, 1, 1, 1] |
| Channel Dims | [64, 128, 256, 512] | 256 | 256 | [256, 128, 64, 64] |
| Specific Settings | Patch Size=4 Adapter Project | Group Norm GRU Update | Heads=4 Expansion=2 | PixelShuffle Global Residual |

## F.1. Architectural Details

The detailed configuration of each component is summarized in Table F.1.

**Context Encoder.** We utilize the FocalNet-L (Large) (Yang et al., 2022) variant pre-trained on ImageNet-22K as our backbone. It employs Focal Modulation to capture long-range dependencies essential for robust global semantic understanding. The backbone consists of 4 hierarchical stages with depths of [2, 2, 18, 2] and an embedding dimension of 192. Since the original FocalNet feature channels ([192, 384, 768, 1536]) differ from our decoder requirements, we employ a set of lightweight $1 \times 1$ convolutions (Adapters) to project them into our optimized working dimensions of [64, 128, 256, 512].

**Micro-Dynamics Decoupler (MDD).** To capture the dynamics of reflection and transmission, we adapt the RAFT architecture (Teed & Deng, 2020). The Event-Motion Encoder consists of parallel feature encoders for both RGB images and Event Voxel Grids, utilizing a 'BasicEncoder' structure (Instance Norm + ReLU). These encoders map inputs to a common feature space at 1/8 resolution with 256 channels. A GRU-based update block iteratively refines the motion field for $M = 12$ iterations. The final output is a 256-channel motion prior tensor.

**Parallax-Attention Rectifier (PAR).** To effectively integrate the dynamics prior (256 channels) into the RGB features (Stage 3, 256 channels), we design a Dual-Channel Block (DCB). Unlike simple concatenation, this module utilizes a Transformer-style architecture to align and refine features:

- **Dual-Stream Self-Attention:** Both RGB and Event branches first pass through independent Multi-Head Self-Attention (MSA) layers to capture global intra-modal dependencies.

- **Bidirectional Cross-Attention:** We employ Cross-Attention layers where contextual features query the dynamics prior. This facilitates the precise transfer of dynamics cues to the structural features.

- **Locally-Enhanced Feed-Forward Network (FFN):** The fused features are processed by a FFN augmented with $3 \times 3$ depth-wise convolutions to enhance local spatial context.

This fusion operates with 4 attention heads and an expansion factor of 2.

**Decoder.** To maintain computational efficiency, our decoder adopts a lightweight design. It consists of 4 stages, where each stage contains only 1 NAFBlock (Chen et al., 2022) followed by a PixelShuffle upsampling layer.

## G. Efficiency-Performance Trade-off Analysis

In this section, we conduct a comprehensive evaluation of the computational efficiency and restoration quality of our proposed EvReflection compared to state-of-the-art baselines. We report the number of parameters (Params), inference runtime, Frames Per Second (FPS), and the PSNR value evaluated on the real-world $EVR^2$ dataset. The quantitative comparisons are summarized in Table G.1.

*Table G.1.* Comparison of computational efficiency and performance. The PSNR results are evaluated on the real-world $EVR^2$ dataset. Runtime and FPS are measured on a single NVIDIA 4090 GPU.

| Metrics | IBCLN | DSRNet | RRW | DSIT | DAI | RDNet | Ours |
|---|---|---|---|---|---|---|---|
| Params (M) ↓ | 44.40 | 144.63 | **27.99** | 326.96 | 1654.18 | 315.89 | 216.73 |
| Runtime (ms) ↓ | 90.0 | 56.2 | 57.2 | 93.2 | 83.2 | 66.8 | **47.7** |
| FPS (Hz) ↑ | 11.1 | 17.8 | 17.5 | 10.7 | 12.0 | 15.0 | **21.0** |
| PSNR (dB) ↑ | 20.59 | 24.35 | 25.05 | 25.65 | 21.25 | 25.97 | **27.25** |

As observed in Table G.1, our model utilizes 216.73 M parameters. We acknowledge that this parameter count is higher than some lightweight baselines. However, a breakdown of the model structure reveals that this size is dominated by the pre-trained backbone, FocalNet (Yang et al., 2022), which accounts for 204 M parameters alone. This implies that our proposed core contributions—the Micro-Dynamics Decoupler (MDD) and Parallax-Attention Rectifier (PAR)—are extremely lightweight, introducing only a marginal overhead of approximately 12.7 M parameters. Therefore, the high total parameter count is a configuration choice for maximizing feature quality rather than an inherent redundancy of our method. In resource-constrained scenarios, the heavy backbone can be readily replaced with lightweight alternatives to significantly reduce the model size without altering the effectiveness of the proposed event-based disentanglement mechanism.

Despite the substantial backbone, EvReflection achieves the fastest inference speed (47.7 ms / 21.0 Hz), significantly outperforming complex RGB-based networks. This efficiency advantage is fundamentally rooted in the superiority of event-based dynamics cues. Motion is the critical discriminator for separating reflection layers. Unlike RGB-based methods that often rely on computationally expensive, iterative optical flow estimation to infer motion from RGB frames, event cameras naturally encode precise micro-dynamics with high temporal resolution. These high-fidelity motion cues are far more effective than complex network architectures. Consequently, our model relieves the burden of heavy motion modeling, allowing for a streamlined, parallelizable feed-forward process that achieves the highest restoration quality (27.25 dB) without the latency bottlenecks of traditional approaches.

