# OpenReview forum: "EvReflection: Event-Driven Micro-Dynamics for Reflection Removal"
_ICML.cc/2026/Conference — ICML 2026 regular_

### Official Review · Reviewer_BvcU · 2026-02-25

**Soundness:** 3
**Presentation:** 3
**Significance:** 3
**Originality:** 3
**Overall Recommendation:** 4
**Confidence:** 4

**Summary:**

This paper proposes EvReflection, an Event-Driven Micro-Dynamics for Reflection Removal method. Specifically, the authors incorporating a Micro Dynamics Decoupler and a Parallax-Attention Rectifier. A physics-based simulation pipeline is designed to address data scarcity and construct the EVR2 benchmark.

**Compliance With Llm Reviewing Policy:**

Affirmed.

**Key Questions For Authors:**

See weakness.

**Limitations:**

The authors do not discuss the limitations. Please add such discussions.

**Strengths And Weaknesses:**

* Strengths
  * The logic of the overall writing is reasonable, and the motivation is strong enough.
  * The overall performance is good and outperforms the other methods.
* Weaknesses
  * Can the proposed method process static scenes? It seems that the events will only triggered in dynamic scenes.
  * Can the proposed method handle video reflection removal problem? Since events can provide temporal information, it is better to show some video results.
  * Please show the qualitative results of DSRNet, RRW in Figure6 and 7. Please give me a reason why these results do not appear in the submitted paper.

---

> ### Author Rebuttal · Authors · 2026-03-31
>
> Thank you for your valuable comments. We address your questions below and will add all the new experiments and discussions to the final paper.
>
> **Q1: Can the proposed method work in static scenes?**
>
> **A1:** Thank you for this question. We tested our model in static scenes where neither the camera nor the objects moved. You can see the results in [Figure A: Static Conditions](https://anonymous.4open.science/r/text-1AB4/figures/static.png).
>
> In these static scenes, event sensors mostly capture random noise. However, our model still performs better than RGB-only methods. The reason should be that our PAR module uses a “matching” tool (cross-attention) to compare RGB images and event data. Since sensor noise is just random and doesn’t match the real shapes in the RGB image, the model automatically ignores the noise. This proves that our network is very reliable and does not purely depend on motion to work correctly.
>
>
> **Q2: Can the proposed method handle the video reflection removal problem?**
>
> **A2:** Thank you for pointing this out. We tested our model on real-world videos and provided the results in this [video link](https://anonymous.4open.science/r/text-1AB4/video/README.md). Here are the details of our test:
>
> * **Data Collection:** It is very hard to record two videos (one with reflections and one without) at the same time. To solve this, we first recorded a video with reflections. Then, we removed the glass and recorded the same scene again, trying to move the camera at the same speed and path. This second video acts as our reference (GT).
> * **Results:** Our model currently processes the video frame by frame. It takes one image and its event data to produce a clean result. As shown in the link (labeled '_in' for input and '_out' for our result), our method successfully removes most reflections in these moving scenes.
> * **Limitations:** Because we process each frame individually, the video flickers slightly. Our current model does not have special tools (like optical flow) to link frames together perfectly. Our main goal here was to show that event data is helpful for this task. We will focus on making the video smoother in our future work.
>
> **Q3: Qualitative comparisons with DSRNet and RRW, and why they were omitted.**
>
> **A3:** Thank you for pointing this out. The omission of DSRNet and RRW from Figures 6 and 7 in the main manuscript was primarily due to strict page limits. To manage space effectively, we prioritized displaying more recent SOTA methods (such as RDNet and DSIT) and a representative multi-frame method (SOLD).
>
> To address your concern and provide a direct comparison, we have now updated the visualizations. The complete qualitative results, which include DSRNet, RRW, and other baselines, are provided in the updated [Figure G: Visual_SIR$^2$](https://anonymous.4open.science/r/text-1AB4/figures/figure6_all.png) and [Figure H: Visual_EVR$^2$](https://anonymous.4open.science/r/text-1AB4/figures/figure7_all.png).
>
> Additionally, please note that the visual results for these earlier methods, including DSRNet and RRW, were actually already included in Appendix Section E of our original submission, which covers comprehensive comparisons across all four datasets for reference.
>
> **Q4: Adding a discussion about the limitations of the proposed method.**
>
> **A4:** Thank you for this suggestion. We agree that adding a discussion on limitations will make our paper more complete and objective. We will add a new "Discussion and Limitations" section to the final version, covering these points:
>
> * **Scenes with No Motion:** Our method relies on movement to separate reflections from the background. If the camera and the objects are perfectly still, the event sensor will only capture random noise, making it difficult to remove reflections.
> * **Extreme Low Light:** Although event sensors are generally better in the dark than normal cameras, very extreme low-light conditions still challenge both the RGB camera and the event sensor.
>
> We will include these details in the final revision to provide a more honest and balanced view of our work.

---

> > ### Author Rebuttal · Reviewer_BvcU · 2026-04-02
> >
> > After reading the rebuttal, I decide to keep my score.
> > Please add the extra experimental results to the camera ready version.

---

> > > ### Author Response · Authors · 2026-04-02
> > >
> > > Dear Reviewer BvcU,
> > >
> > > Thank you for your time reviewing our paper and for your positive feedback! We are glad that our rebuttal has successfully addressed your concerns. We will certainly include all the additional experimental results in the camera-ready version.
> > >
> > > Best regards,
> > >
> > > The Authors

---

### Official Review · Reviewer_VTWj · 2026-03-03

**Soundness:** 3
**Presentation:** 3
**Significance:** 4
**Originality:** 4
**Overall Recommendation:** 5
**Confidence:** 4

**Summary:**

The authors tackle the problem of removing reflections from a single image by introducing event cameras. The main idea is that subtle camera shakes produce different event signals for the background and the reflection layers because they are at different depths. To implement this, the paper proposes the EvReflection network. It uses a Micro-Dynamics Decoupler (MDD) to learn motion features from events and a Parallax-Attention Rectifier (PAR) to remove reflections from RGB features. A new real-world dataset named EVR² and a synthetic data pipeline are also provided to train and test the model.

**Compliance With Llm Reviewing Policy:**

Affirmed.

**Final Justification:**

I appreciate the authors' time and effort in preparing this rebuttal. I have also read the comments from other reviewers and the corresponding rebuttals. The supplementary experimental results are satisfactory, especially the new visual comparisons.

Figure F clearly demonstrates the advantage of the method. Additionally, the new table showing the performance of lighter backbones like ResNet18 perfectly addresses my concern about model complexity.

Since all my concerns are fully addressed, I am happy to raise my score from 4 (Weak Accept) to 5 (Accept).

**Key Questions For Authors:**

1. The parameter count is dominated by FocalNet. Are the proposed MDD and PAR modules plug-and-play? Could you provide a small experiment showing their performance when combined with a lightweight backbone (e.g., a simple ResNet or small CNN)?
2. Could you please provide some visual comparisons between EvReflection and the event-based baselines (EvLight, DeblurSR, EFNet) in the rebuttal PDF?
3. Did you try adding any intermediate loss to supervise the motion output of the MDD module? Why are only RGB-level losses sufficient for learning accurate event dynamics?

**Limitations:**

The paper handles limitations briefly. The authors should expand the discussion to include the reliance on heavy backbones for high performance, making the paper more solid and objective.

**Strengths And Weaknesses:**

Pros：
+ The motivation is highly practical. Using the micro-second temporal resolution of event cameras to find motion parallax is a natural and effective way to separate image layers.
+ The dataset construction is impressive. The EVR² dataset is carefully divided by glass thickness (3mm, 5mm, 8mm). This provides a very clear metric for how well the method handles different ghosting levels.
+ The experimental results are strong. The proposed method clearly outperforms previous RGB-based single-image and multi-image baselines on both synthetic and real data.

Cons:
- Model Size and Complexity: As shown in Appendix G, the model has over 216M parameters, and about 204M of them come from the FocalNet backbone. This makes the network quite heavy. The paper does not show whether the MDD and PAR modules work well with lighter backbones.
- Missing Visual Comparisons: In Table 4, the authors quantitatively compare their model with other event-based low-level vision networks (like EvLight and DeblurSR). However, these event-based baselines lack qualitative (visual) comparisons, making it hard to see which specific artifacts the competitors produce.
- Loss Function Design: The network is trained using standard RGB reconstruction and perceptual losses. Since the MDD extracts dynamic features from events, it is surprising that no event-domain or motion-domain loss is applied to directly supervise the MDD module.

---

> ### Author Rebuttal · Authors · 2026-03-31
>
> We really appreciate your helpful feedback. We have answered all your questions below and will include these new experiments and discussions in the final version of our paper.
>
> **Q1: Model size and performance with smaller backbones.**
>
> **A1:** Our MDD and PAR modules are flexible and can work with different backbones. To address your concern, we tested our modules with two smaller backbones: (a) a smaller version of FocalNet and (b) a standard, lightweight ResNet18 on three synthetic datasets and our collected real dataset EVR². The results are shown below:
>
> | Backbone | Params | Object | Postcard | Wild | EVR² |
> | :---: | :---: | :---: | :---: | :---: | :---: |
> | (a) FocalNetL64 | 23.6M | 28.18/0.947 | 26.87/0.934 | 29.31/0.915 | 26.65/0.881 |
> | (b) ResNet18 | 11.5M | 28.25/0.945 | 26.93/0.934 | 28.70/0.922 | 26.20/0.871 |
> | **(c) FocalNetL (Ours)** | **204M** | **28.63/0.947** | **28.57/0.946** | **30.07/0.927** | **27.25/0.896** |
>
> The results show that even when the model is nearly 18x smaller (from 204M to 11.5M), the performance drop is very small (less than 2.0 dB). On the Object dataset, the drop is even less than 0.5 dB. This proves that our MDD and PAR modules are very efficient. Even the smallest version (ResNet18) still performs better than previous RGB-only methods.
>
> **Q2: Visual comparisons with other event-based models.**
>
> **A2:** Thank you for this suggestion. We have added a visual comparison with other event-based models (EvLight, DeblurSR, and EFNet) in [Figure F: Other Event](https://anonymous.4open.science/r/text-1AB4/figures/event-based.png).
>
> Most event-based models think that all motion signals are important details, so they often make reflections even stronger. As shown in the figure, while other methods leave "ghost" images or blurry spots, our method successfully separates the layers. This produces much cleaner results that match the ground truth (GT) best.
>
> **Q3: Why do we not use event-domain or motion-domain losses for the MDD module?**
>
> **A3:** Thanks for the great point. We chose not to use extra motion losses for a few reasons:
>
> * **Focus on the final image:** Our main goal is to get a clean final image. The RGB loss is already very strong. To make the picture look good, the model naturally learns to find the right motion signals through the MDD module.
> * **Built-in motion knowledge:** The MDD module is based on a famous motion-tracking network [1]. Since it starts with pre-trained weights, it already knows how to handle motion from the beginning.
> * **Great results:** Our current method already gets the best results (SOTA). Since it works so well, we kept it simple for now. We will look into more ways to guide the model in our future work.
>
> **References:**
> [1] Wan et al., "Learning dense and continuous optical flow from an event camera," TIP 2022.
>
> **Q4: Expanding the limitations section, especially regarding large backbones.**
>
> **A4:** Thanks for the great advice. We agree that a better discussion on limitations will make our paper more honest and solid. We will add a new "Discussion and Limitations" section to the final version, focusing on these points:
>
> * **Model Size:** As you mentioned, our best results currently need a very large backbone (FocalNet). Although our modules still work well with smaller backbones (like ResNet18), the large version is quite heavy. We will discuss the need for making the model smaller and faster in the future.
> * **No-Motion Scenes:** Our method uses motion to separate the two layers. If the camera and the scene are perfectly still, the event sensor will only see noise. This makes it very hard to remove reflections in those cases.
> * **Extreme Low Light:** Even though events are great in the dark, very extreme low-light conditions still challenge both the camera and the event sensor.
>
> We will include these details to give a more complete and objective view of our work.

---

> > ### Author Rebuttal · Reviewer_VTWj · 2026-04-02
> >
> > I appreciate the authors' time and effort in preparing this rebuttal. I have also read the comments from other reviewers and the corresponding rebuttals. The supplementary experimental results are satisfactory, especially the new visual comparisons.
> >
> > Figure F clearly demonstrates the advantage of the method. Additionally, the new table showing the performance of lighter backbones like ResNet18 perfectly addresses my concern about model complexity.
> >
> > Since all my concerns are fully addressed, I am happy to raise my score from 4 (Weak Accept) to 5 (Accept).

---

> > > ### Author Response · Authors · 2026-04-02
> > >
> > > Dear Reviewer VTWj,
> > >
> > > Thank you for your time and for upgrading your score! We are glad the new visual comparisons and experiments successfully addressed your concerns. We will include these additional results in the camera-ready version.
> > >
> > > Best regards,
> > >
> > > The Authors

---

### Official Review · Reviewer_MVM8 · 2026-03-12

**Soundness:** 3
**Presentation:** 3
**Significance:** 3
**Originality:** 3
**Overall Recommendation:** 4
**Confidence:** 5

**Summary:**

This paper introduces EvReflection, a novel event-driven reflection removal network that leverages dynamic cues for effective layer separation. The authors construct both a simulated dataset and a real-world dataset for evaluation. Extensive experiments demonstrate that EvReflection outperforms existing methods, particularly in challenging scenarios.

**Compliance With Llm Reviewing Policy:**

Affirmed.

**Final Justification:**

After reading the author’s response, all my concerns have been addressed, so I’m keeping my original positive score.

**Key Questions For Authors:**

1. For the real-world dataset, how was the ground truth obtained? The authors should provide a clear and detailed description of the data collection and labeling process. Transparency in ground truth generation is essential for assessing the validity of the dataset and the reliability of the quantitative results.

2. The primary advantages of event cameras are their ability to capture high-speed motion and operate in extreme lighting conditions. To truly demonstrate the value of the proposed event-guided approach, the authors should provide qualitative results (and ideally quantitative evaluations) in such challenging scenarios. For example, could the authors show visual comparisons of reflection removal in high-speed dynamic scenes or low-light conditions? This would more convincingly justify the use of event data for this task.

3. I commend the authors for including a comparison with event-based baselines in Section 4.5. However, this section raises an important question: Could classical hybrid restoration frameworks that jointly process events and frames also be applied to event-guided image reflection removal? In principle, this is a strongly supervised task; given sufficient dataset quality and labeled data, such frameworks might achieve good results. The authors currently argue that these methods "lack specific mechanisms to decouple two additive image layers (reflection and transmission)," but this may not be the fundamental limitation. I would appreciate a more detailed explanation of why existing hybrid frameworks are unsuitable for this specific task, beyond the lack of explicit decoupling mechanisms.

**Limitations:**

Please the authors give a short discussion on the limitation in camera-ready version.

Additionally, please the authors cite the omitted some references:

1. For example, regarding physical-based simulation, please refer to and cite [1].

2. We hope the authors can discuss whether reflection removal is beneficial for downstream analysis tasks, such as object detection [2] and depth estimation [3], which leverage events and frames for downstream tasks.

[1] Physical-based event camera simulator, ECCV 2024.

[2] SODFormer: Streaming object detection with transformer using events and frames, TPAMI 2023.

[3] High-rate monocular depth estimation via cross frame-rate collaboration of frames and events, IJCV 2025.

**Strengths And Weaknesses:**

Strengths:

1. Event-guided image reflection removal is a highly novel and meaningful research direction. The authors have identified an important problem at the intersection of event-based vision and computational photography.

2. The paper is exceptionally well-written. The authors demonstrate strong expository skills, from the clarity of the writing to the quality of the figures and the precision of the mathematical formulations. These elements collectively make the work highly accessible and professionally presented.

Weaknesses:
1. Some descriptions in the manuscript slightly overstate the novelty or contribution, particularly in less critical sections. I would encourage the authors to adopt more rigorous and measured language throughout. For example, the term "physical-based simulation" is used, but the simulation does not appear to model actual light transport or optical pathways. I would caution against using the term "physical-based" unless true physical optics are simulated. Similarly, the authors refer to their simulated and real-world data as "benchmarks." Unless these datasets are intended to serve as a standard evaluation suite for the community, it is more accurate and modest to simply refer to them as "datasets."

2. The topic of event-guided image reflection removal is indeed intriguing. However, reflection is inherently a lens-related phenomenon and is not directly tied to the camera sensor itself. This raises a fundamental question: why would an event camera be particularly suited for reflection removal? If a standalone event camera also captures reflections in its output, how would one remove them? To convincingly demonstrate that events genuinely aid in removing reflections from images, the authors need to provide solid theoretical analysis and experimental evidence. For instance, could the authors show results from the event camera alone, the RGB image alone, and the combined approach? A clear ablation study is necessary to isolate and validate the contribution of the event stream.

3. The primary advantages of event cameras lie in high-speed imaging and low-light scenarios. However, the datasets constructed in this work do not appear to fully showcase these capabilities, nor are these extreme conditions emphasized or validated in the experiments. To truly demonstrate the value of an event-guided approach, the authors should consider evaluating their method in high-speed dynamic scenes or challenging low-light conditions where traditional RGB-based methods typically fail. This would provide a more compelling justification for introducing event data into the reflection removal task.

---

> ### Author Rebuttal · Authors · 2026-03-31
>
> Thank you for your helpful feedback. Here are our responses. We will add all improvements to the final version of our paper.
>
> **Q1: Refine some overstated descriptions.**
>
> **A1:** Thank you for your helpful comments. We will follow your advice and make the following changes in our final revision:
>
> 1) Regarding the simulation, we will replace the term "physical-based simulation" with "parallax-aware simulation". This better shows that our focus is on motion parallax between layers.
> 2) Regarding the data, we will change the term "benchmark" to "benchmark dataset". Since our EVR² is the first to provide well-aligned event-RGB data and ground truths for the new task of event-based reflection removal, we hope it can help future research while staying modest.
>
> **Q2: Why event cameras for reflection removal and evidence of their contribution?**
>
> **A2:** Thank you for these important questions. We agree that reflection is a lens-related problem. Here is our explanation and experimental evidence to show why events are helpful.
>
> * **Theory:** Layers at different depths have different motion speeds (parallax). Standard cameras are too slow to catch this, but event cameras are fast enough. We provide detailed theoretical derivations in Section 3.1 and Appendix A & B.
> * **Results:** We compared three setups below (PSNR↑/SSIM↑):
>
> | Configurations | Object | Postcard | Wild | EVR² |
> | :---: | :---: | :---: | :---: | :---: |
> | (a) Event-only | 26.63/0.883 | 26.68/0.944 | 26.29/0.880 | 25.86/0.871 |
> | (b) RGB-only | 27.34/0.938 | 27.09/0.958 | 26.68/0.902 | 26.26/0.885 |
> | **(c) Ours** | **28.63/0.947** | **28.57/0.946** | **30.07/0.927** | **27.25/0.896** |
>
> RGB-only (b) fails to separate mixed layers, and Event-only (a) lacks color. Our combined approach (c) performs best by using both color and motion.
>
> **Q3: Performance in high-speed and low-light scenes.**
>
> **A3:** Thank you for this suggestion. To show the benefits of events, we tested our method in low-light and high-speed scenes. As shown in in [Figure B: Low-Light](https://anonymous.4open.science/r/text-1AB4/figures/lowlight.png) and [Figure C: High-Speed](https://anonymous.4open.science/r/text-1AB4/figures/high-speed.png), RGB images lose details or become blurry in these cases, causing standard methods to fail. However, our method uses high-quality event signals to recover clear edges and remove reflections successfully. These results prove that our approach is much more robust than RGB-only methods in tough conditions.
>
>
> **Q4: Details on ground truth collection for the real-world dataset.**
>
> **A4:** To get high-quality data, we used a "stop-and-go" method. Here is how we collected the images and events:
>
> 1. **Recording Events:** We moved the camera on a straight track (linear slide) to record the event data during the movement.
> 2. **Capturing Images:** We stopped the camera at a specific point on the track. At this same spot, we took two photos:
>    - One photo with the glass to get the input image.
>    - One photo after removing the glass to get the clean GT image.
>
> Because the camera stayed in the exact same position for both photos, the input and the GT match perfectly. Since the scene did not change, these images also match the event data recorded at that position. This method ensures our data is reliable and accurate.
>
> **Q5: Why are existing hybrid event-RGB models unsuitable for reflection removal?**
>
> **A5:** Thank you for this question. Most existing event-RGB models for deblurring or super-resolution are designed to add details from events to the image. However, reflection removal is different because it requires the model to separate or remove one layer from another.
>
> Even though a standard model could theoretically learn to "subtract" the reflection, it is very hard for it to find exactly what to delete across many channels without hurting the background. Without a special tool like our PAR gating mechanism, the network does not have a clear way to block unwanted signals. This makes the training process very difficult, and the model often fails to separate the two layers well. Our method is better because it is built specifically to handle this separation task.
>
> **Q6: Citations, limitations, and value for downstream tasks.**
>
> **A6:** Thank you for the suggestions. We will add the missing references and a limitations section in the final version. We also tested our method with depth estimation and object detection. As shown in [Figure D: Depth](https://anonymous.4open.science/r/text-1AB4/figures/depth_estimation.png) and [Figure E: Detection](https://anonymous.4open.science/r/text-1AB4/figures/object_detection.png), reflections cause "ghost" images that confuse these models. Our method removes these reflections, which helps the models produce much more accurate depth maps and detection results.

---

### Official Review · Reviewer_VkZw · 2026-03-12

**Soundness:** 3
**Presentation:** 3
**Significance:** 3
**Originality:** 3
**Overall Recommendation:** 5
**Confidence:** 5

**Summary:**

This paper proposes EvReflection, an event-driven network for single-image reflection removal. Traditional RGB-based methods suffer from severe ill-posedness because they struggle to separate reflection and transmission layers from a single static frame. To address this, the authors use event cameras to capture micro-motions (e.g., slight hand tremors), which naturally provide motion parallax cues. The paper introduces a Micro-Dynamics Decoupler (MDD) to extract motion priors from event voxel grids, and a Parallax-Attention Rectifier (PAR) to fuse these priors with RGB features. To solve the lack of paired training data, the authors design a physics-based simulator and collect a new real-world benchmark named EVR², which contains different glass thicknesses.

**Compliance With Llm Reviewing Policy:**

Affirmed.

**Final Justification:**

Thanks to the authors for the detailed response and new experiments, which fully address my initial concerns. Please include the additional results provided in the rebuttal into the camera-ready version. I am maintaining my positive rating.

**Key Questions For Authors:**

1. How does the EvReflection framework perform when the camera is strictly static? Does the lack of valid event dynamics cause performance drops compared to RGB-only models?
2. In Appendix D, the temporal bin size of the event voxel grid is set to B=20. How sensitive is the final decoupling performance to this hyperparameter? A brief ablation study on B would be helpful.
3. In very dark scenes where the event stream is noisy, how does the cross-attention mechanism in the PAR module prevent sensor noise from corrupting the RGB features?

**Limitations:**

The authors provide a strong theoretical foundation. However, the limitation discussion is somewhat weak. The paper should explicitly discuss the failure cases or boundary conditions, such as strictly zero-motion scenarios (e.g., tripod setups) and extreme low-light environments where event noise dominates.

**Strengths And Weaknesses:**

**Strengths:**
* **Originality & Soundness:** The motivation is physically solid. Using event cameras to break the ill-posedness of single-image reflection removal is a novel and effective idea. The mathematical proofs in the appendix are detailed and convincing, showing how the event-based gradient constraint guarantees a unique solution. The dual-branch design in the MDD module perfectly matches this physical model.
* **Significance:** The EVR² benchmark is a valuable contribution to the low-level vision community. Testing the model on different glass thicknesses (3mm, 5mm, 8mm) is a rigorous way to evaluate double-reflection and ghosting artifacts in real-world scenarios.
* **Presentation:** The paper is well-organized. The logical flow from physical formulation to deep learning network design is clear and easy to follow. The experimental comparisons are comprehensive.

**Weaknesses:**
* **Boundary Conditions (Soundness):** The proposed method strongly relies on micro-motions. The paper lacks a discussion on extreme static conditions. For example, if the camera is fixed on a tripod, the event stream will mostly contain sensor noise. It is unclear if the network will degrade to a standard RGB baseline or produce new artifacts due to the lack of dynamic priors.
* **Noise Robustness (Significance):** Event cameras have low signal-to-noise ratios in extreme low-light environments. The paper does not provide enough analysis on how the PAR module handles pure noise when structural edges are missing in the event stream.

---

> ### Author Rebuttal · Authors · 2026-03-31
>
> We sincerely thank you for your constructive feedback. Below are our detailed point-by-point responses to your concerns. We will incorporate all the supplemented experiments and discussions into the final revision.
>
> **Q1:** **Robustness under static and low-light conditions.**
>
> **A1:** Thank you for highlighting these critical scenarios. We have supplemented experiments under strictly static camera setups and extreme low-light environments, comparing EvReflection with recent SOTA methods.
>
> As shown in [Figure A: Static Conditions](https://anonymous.4open.science/r/text-1AB4/figures/static.png), when there is no camera motion, the event stream is dominated by inherent sensor noise. However, the results demonstrate that our network avoids introducing artifacts and maintains a robust performance comparable to standard RGB-only baselines, with no significant drops. We attribute this noise-filtering capability to our PAR module. Since random sensor noise lacks spatial correlation with the physical structures in the RGB image, the cross-attention mechanism naturally assigns negligible weights to these uncorrelated features. This allows the model to safely bypass the noise and fall back on RGB priors in both strictly static and noisy dark scenes.
>
> Furthermore, [Figure B: Low-light Environments](https://anonymous.4open.science/r/text-1AB4/figures/lowlight.png) shows that our method achieves superior performance even under severe low-light conditions. We attribute this to the high dynamic range (HDR) characteristic of event sensors. While standard RGB frames suffer from severe degradation and loss of structural edges in the dark, event cameras can still reliably capture micro-dynamics. Our MDD module effectively extracts these faint but valid motion priors, successfully compensating for the degraded RGB modality.
>
> **Q2:** **Sensitivity analysis of the temporal bin size $B$.**
>
> **A2:** Thank you for this constructive suggestion. We conducted a quantitative ablation study varying $B$ ∈ {5, 10, 20, 30} across three synthetic datasets (Object, Postcard, Wild) and our real-world EVR² dataset. The results (PSNR↑ / SSIM↑) are summarized below:
>
> | # Bins | Object | Postcard | Wild | EVR² |
> | :---: | :---: | :---: | :---: | :---: |
> | 5 | 28.46 / 0.947 | 27.71 / 0.941 | 28.31 / 0.915 | 26.33 / 0.887 |
> | 10 | 28.59 / 0.946 | 28.14 / 0.939 | 28.98 / 0.918 | 27.11 / 0.895 |
> | **20 (Ours)** | **28.63 / 0.947** | **28.57 / 0.946** | **30.07 / 0.927** | **27.27 / 0.896** |
> | 30 | 28.58 / 0.947 | 28.33 / 0.943 | 29.67 / 0.922 | 27.21 / 0.896 |
>
> As shown in the table, the model achieves the best overall performance at $B=20$, validating our hyperparameter choice. We believe this trend stems from the impact of temporal granularity on feature entanglement. Specifically, a small $B$ lacks sufficient resolution, leading to motion trails and structural edges overlapping within the same voxel bins. Setting $B=20$ provides adequate temporal granularity to cleanly separate these micro-dynamics, while larger values yield diminishing returns as the motion is already fully resolved.
>
>
> **Q3:** **Add more discussion about limitations.**
>
> **A3:** Thank you for this constructive suggestion. We briefly discuss the limitations of our work here and will add more details in the revised version:
>
> 1) Since EvReflection relies on motion parallax, the event stream inevitably degrades into sensor noise in strictly zero-motion scenarios (e.g., tripod setups).
> 2) Both RGB and event modalities suffer from significant signal loss in extremely low-light environments where event noise dominates.
> 3) The current high-performance backbone introduces non-negligible computational overhead, which we plan to address with lightweight designs in future work.

---

> > ### Author Rebuttal · Reviewer_VkZw · 2026-04-04
> >
> > Thanks to the authors for the detailed response and new experiments, which fully address my initial concerns. Please include the additional results provided in the rebuttal into the camera-ready version. I am maintaining my positive rating.

---

> > > ### Author Response · Authors · 2026-04-04
> > >
> > > Dear Reviewer VkZw,
> > >
> > > Thank you for your time on our paper and your continued positive rating. We will integrate all the new experiments and details from the rebuttal into the final manuscript.
> > >
> > > Best regards,
> > >
> > > The Authors

---

### Decision · Program_Chairs · 2026-04-30

**Decision:**

Accept (regular)

**Comment:**

This paper proposes an event-guided method for single-image reflection removal. The reviewers found the idea novel and well motivated. In particular, they appreciated the use of event signals to capture micro-dynamics and motion parallax cues that help separate reflection and background layers. The reviewers also found the technical design clear and the experimental results strong. The new real-world dataset is another valuable part of the paper.

The main concerns were about static scenes, low-light conditions, model complexity, and the need for clearer discussion of limitations and additional comparisons. In the rebuttal, the authors addressed these points well with new experiments, visual results, and clearer explanations. The additional results on static scenes, low-light settings, lighter backbones, and event-based baselines strengthened the paper further.

Overall, AC finds that this paper makes a meaningful contribution to reflection removal and event-based vision. The method is technically sound, the evaluation is strong, and the rebuttal addressed the main concerns. AC therefore recommends acceptance.